# TRPC5 controls the adrenaline-mediated counter regulation of hypoglycemia

Jenny Bröker-Lai[1,2,13], José Rego Terol[3,13], Christin Richter[1,14], Ilka Mathar[1,2,14], Angela Wirth[1,2,14], Stefan Kopf[4,5,14], Ana Moreno-Pérez [ID][3], Michael Büttner[6], Linette Liqi Tan [ID][1], Mazen Makke[3], Gernot Poschet [ID][6], Julia Hermann [ID][1], Volodymyr Tsvilovskyy[1,2], Uwe Haberkorn[7], Philipp Wartenberg[8], Sebastian Susperreguy[9], Michael Berlin [ID][1,2], Roger Ottenheijm [ID][1,2], Koenraad Philippaert [ID][1,2], Moya Wu[5], Tobias Wiedemann [ID][5], Stephan Herzig[5], Anouar Belkacemi [ID][1,2], Rebecca T Levinson [ID][10], Nitin Agarwal[1], Juan E Camacho Londoño[1,2], Bert Klebl[11], Klaus Dinkel[11], Frank Zufall [ID][3], Peter Nussbaumer [ID][11], Ulrich Boehm [ID][8], Rüdiger Hell [ID][6], Peter Nawroth [ID][4,12], Lutz Birnbaumer[9], Trese Leinders-Zufall [ID][3], Rohini Kuner [ID][1], Markus Zorn[4], Dieter Bruns[3], Yvonne Schwarz[3,15✉] & Marc Freichel [ID][1,2,15✉]

## Abstract

Hypoglycemia triggers autonomic and endocrine counter-regulatory responses to restore glucose homeostasis, a response that is impaired in patients with diabetes and its long-term complication hypoglycemia-associated autonomic failure (HAAF). We show that insulin-evoked hypoglycemia is severely aggravated in mice lacking the cation channel proteins TRPC1, TRPC4, TRPC5, and TRPC6, which cannot be explained by alterations in glucagon or glucocorticoid action. By using various TRPC compound knock-out mouse lines, we pinpointed the failure in sympathetic counter-regulation to the lack of the TRPC5 channel subtype in adrenal chromaffin cells, which prevents proper adrenaline rise in blood plasma. Using electrophysiological analyses, we delineate a previously unknown signaling pathway in which stimulation of PAC1 or muscarinic receptors activates TRPC5 channels in a phospholipase-C-dependent manner to induce sustained adrenaline secretion as a crucial step in the sympathetic counter response to insulin-induced hypoglycemia. By comparing metabolites in the plasma, we identified reduced taurine levels after hypoglycemia induction as a commonality in TRPC5-deficient mice and HAAF patients.

**Keywords** TRPC5 Channels; Chromaffin Cells; Adrenaline Secretion; Hypoglycemia Associated Autonomic Failure; Calcium Signaling
**Subject Categories** Genetics, Gene Therapy & Genetic Disease; Metabolism; Neuroscience

## Introduction

Diabetes mellitus is a global pandemic leading to micro- and macrovascular long-term complications and neuropathies (Zheng et al, 2018). The latter includes defects in the modulation of energy balance by the autonomous nervous system that are described in patients with type 1 and type 2 diabetes as hypoglycemia-associated autonomic failure (HAAF). In HAAF, the counter-regulatory responses to low blood glucose concentrations are impaired. When hypoglycemic episodes occur repeatedly, the blood glucose threshold for initiating metabolic counter-regulation is lowered and, thus, the re-establishment of euglycemia, is retarded in these patients (Cryer, 2013; Senthilkumaran et al, 2016). Consequently, HAAF patients experience harmful hypoglycemic episodes with progressively higher frequency and longer durations—a vicious cycle of continuously aggravating HAAF pathology. No specific treatments or diagnostic tools, which allow to predict the susceptibility for HAAF episodes, are currently available, and patient management is restricted to the avoidance of hypoglycemic events.

Although insulin and glucagon are also able to act as counter-regulatory hormones following hypoglycemia, it is mostly the adrenaline-driven mechanisms that restore normal glucose levels

[1]Institute of Pharmacology, Heidelberg University, Heidelberg, Germany. [2]DZHK (German Centre for Cardiovascular Research), partner site Heidelberg/Mannheim, Heidelberg, Germany. [3]Center for Integrative Physiology and Molecular Medicine (CIPMM), Saarland University, Homburg, Germany. [4]Klinik für Endokrinologie, Diabetologie, Stoffwechsel und Klinische Chemie, Heidelberg, Germany. [5]Institute for Diabetes and Cancer, Helmholtz Diabetes Center, Helmholtz Zentrum Muenchen, German Research Center for Environmental Health (GmbH), 85764 Neuherberg, Germany. [6]Metabolomics Core Technology Platform, Centre for Organismal Studies Heidelberg (COS Heidelberg), Heidelberg, Germany. [7]Nuclear Medicine, Heidelberg University Hospital, Heidelberg, Germany. [8]Experimental and Clinical Pharmacology and Toxicology, Center for Molecular Signaling (PZMS), Saarland University, Homburg, Germany. [9]Signal Transduction Laboratory, Institute of Biomedical Research (BIOMED UCA CONICET) Edificio San José, Piso 3 School of Biomedical Sciences, Pontifical Catholic University of Argentina, Buenos Aires, Argentina. [10]Department of General Internal Medicine and Psychosomatics, Heidelberg University Hospital, Heidelberg, Germany. [11]Lead Discovery Center GmbH, Dortmund, Germany. [12]Deutsches Zentrum für Diabetesforschung (DZD e.V), München-Neuherberg, Germany. [13]These authors contributed equally as first authors: Jenny Bröker-Lai, José Rego Terol. [14]These authors contributed equally: Christin Richter, Ilka Mathar, Angela Wirth, Stefan Kopf. [15]These authors jointly supervised this work: Yvonne Schwarz, Marc Freichel. ✉E-mail: Yvonne.Schwarz@uks.eu; Marc.Freichel@pharma.uni-heidelberg.de

during hypoglycemic events in patients with diabetes (Cryer, 2005). Furthermore, in patients with HAAF complications, the sympathetically stimulated release and production of adrenaline are blunted (LaGamma et al, 2014). To date, there has been a lack of in-depth understanding of the impaired autonomic response to hypoglycemia that develops in diabetes (Senthilkumaran et al, 2016).

The first step of counter-regulation is the sensing of hypoglycemia as a form of metabolic stress by specialized hypothalamic neurons which project directly to sympathetic preganglionic neurons in the intermediolateral nucleus (IML) of the spinal cord called splanchnic nerves (Verberne et al, 2014). Hypothalamic neurons transmit information also to catecholaminergic neurons in the hindbrain's rostral ventrolateral medulla (RVLM), which contribute to glucose sensing and send signals to the sympathetic nerves in the spinal cord (Routh et al, 2012; Verberne et al, 2014). Parallel to the hypothalamus, glucose sensing also takes place directly in the RVLM neurons. The sympathetic splanchnic nerve in the spinal cord eventually sends its axons to the catecholaminergic chromaffin cells in the adrenal medulla to stimulate the sustained release of adrenaline via the neurotransmitters acetylcholine (ACh) and pituitary adenylate cyclase-activating polypeptide (PACAP) (Carbone et al, 2019; Smith and Eiden, 2012). Whereas signal transduction evoked by ACh in chromaffin cells via the ionotropic nicotinic (nAChR) acetylcholine receptors and subsequent induction of catecholamine secretion is well described (Wu et al, 2010), the underlying signaling events downstream of the stimulation of muscarinic (M1) and the PACAP receptor (PAC1) that determine sustained catecholamine secretion are still only sparsely understood. 2-APB-sensitive cation currents with similarities to currents conducted by TRPC channels were proposed as mediators of PACAP-regulated long-term catecholamine secretion from chromaffin cells (Mustafa et al, 2007; Smith and Eiden, 2012).

TRPC channels (TRPC1–TRPC7) are structurally homologous members of the family of TRP cation channels. By mediating $Na^+$- and $Ca^{2+}$ influx, TRPC cation channels contribute to membrane depolarization and induce $Ca^{2+}$-dependent intracellular processes such as metabolic signaling in neurons. Cation entry through TRPC5 channels, for instance, has been reported to control the insulin-, leptin-, and serotonin-evoked excitations of hypothalamic anorexigenic POMC neurons, thereby contributing to the neurogenic regulation of energy balance (Gao et al, 2017; Qiu et al, 2018; Sohn et al, 2011). In hippocampal neurons, TRPC5 regulates presynaptic $Ca^{2+}$ dynamics and, in turn, synaptic strength (Schwarz et al, 2019). TRPC4 is essential for GABA release from thalamic interneurons (Munsch et al, 2003) and can support exocytosis in neuroendocrine PC12 cells (Obukhov and Nowycky, 2002). In primary chromaffin cells, the endpoints of the autonomous nervous system, TRPC1 (Marom et al, 2011) and TRPC3 (Liu et al, 2017) have been shown to contribute to the signaling leading to catecholamine release when the cells are stimulated by bradykinin or histamine and angiotensin II, respectively.

Here, we show that diabetic mice lacking TRPC1, 4, 5, and 6 exhibit increased mortality under regular insulin treatment. Using a combination of TRPC-deficient mouse models and pharmacological tools, we unveiled that functional loss of TRPC5 in catecholaminergic cells leads to a defective counter-regulation to insulin-evoked hypoglycemia due to lack of increase in plasma adrenaline, thereby disabling the re-establishment of euglycemia. Our experiments show that stimulation of chromaffin cells by muscarine or the neuropeptide PACAP engages in a phospholipase-C-dependent (PLC) manner TRPC5 channels as common effector molecule to drive sustained catecholamine release.

The comparative metabolome analysis of the plasma from TRPC5-deficient mice and HAAF patients during hypoglycemia pinpointed commonalities in the metabolic signature. Collectively, these results assign TRPC5 channels a crucial role in adrenaline secretion and the stress response to hypoglycemia. They implicate TRPC5 channels as potential targets for diagnostic and pharmacological intervention in patients with diabetes and HAAF complications.

# Results

## Trpc1/4/5/6^{-/-} mice develop aggravated hypoglycemia following insulin treatment

To investigate the role of TRPC channels in long-term diabetic complications, we induced diabetes in mice using a streptozotocin protocol where chronic hyperglycemia was controlled by applications of long-acting insulin (Sachdeva et al, 2018) (Fig. 1A). After 2 weeks of insulin treatment, 50% of $Trpc1/4/5/6^{-/-}$ animals died, whereas all wild-type mice survived the entire period of 20 weeks of insulin treatment (Fig. 1B). When we lowered the insulin dose for the $Trpc1/4/5/6^{-/-}$ animals after two weeks (see methods), their mortality rate could be largely reduced (Fig. 1B). To test whether diabetic complications or the insulin treatment caused this phenomenon, we performed an insulin tolerance test (ITT) on naive (non-diabetic) $Trpc1/4/5/6^{-/-}$ animals (Fig. 1C,D). Indeed, compared with the wild-type mice, the TRPC1/4/5/6-deficient mice exhibited aggravated hypoglycemia at any time point analyzed until 60 min after the insulin administration.

When TRPC1/4/5/6-deficient animals were subjected to a glucose tolerance test (GTT), the blood glucose levels of the mutant and wild-type animals ascended and subsequently descended in a similar fashion, suggesting an unchanged clearance of glucose from the blood in $Trpc1/4/5/6^{-/-}$ mice (Fig. 1F,G). As the glucose concentration in the blood is affected by multiple regulatory circuits and mechanisms (Cryer, 2006), we also analyzed the insulin-mediated glucose utilization in vivo. For this, the radioactively labeled glucose derivative $^{18}$F-fluorodeoxyglucose ($^{18}$F-FDG) was administered to the $Trpc1/4/5/6^{-/-}$ animals and quantified in the heart, kidney, brain, and fat through positron emission tomography (PET) (Fig. 1E). The comparable uptake of $^{18}$F-FDG into these tissues of both genotypes showed that glucose uptake in insulin-sensitive tissues is not changed in $Trpc1/4/5/6^{-/-}$ mice, indicating that TRPC deletion does not enhance insulin sensitivity. Hypoglycemia provokes elevated plasma catecholamine levels triggered via the hypothalamo-sympatho-adrenal (HSA) axis, thereby inducing gluconeogenesis and glycogenolysis. Consequently, the insulin application raised adrenaline in wild-type controls compared to the saline-treated group (Fig. 1H). In sharp contrast, adrenaline plasma levels were significantly lowered in $Trpc1/4/5/6^{-/-}$ animals. Collectively, these results show that $Trpc1/4/5/6^{-/-}$ mice develop an aggravated insulin-induced hypoglycemia which is associated with reduced plasma adrenaline levels.

## Hypoglycemia counter regulation depends on TRPC5 channels

Similar defects in glucose homeostasis (Appendix Fig. S1A) and plasma adrenaline rise (Appendix Fig. S1B) were observed in the

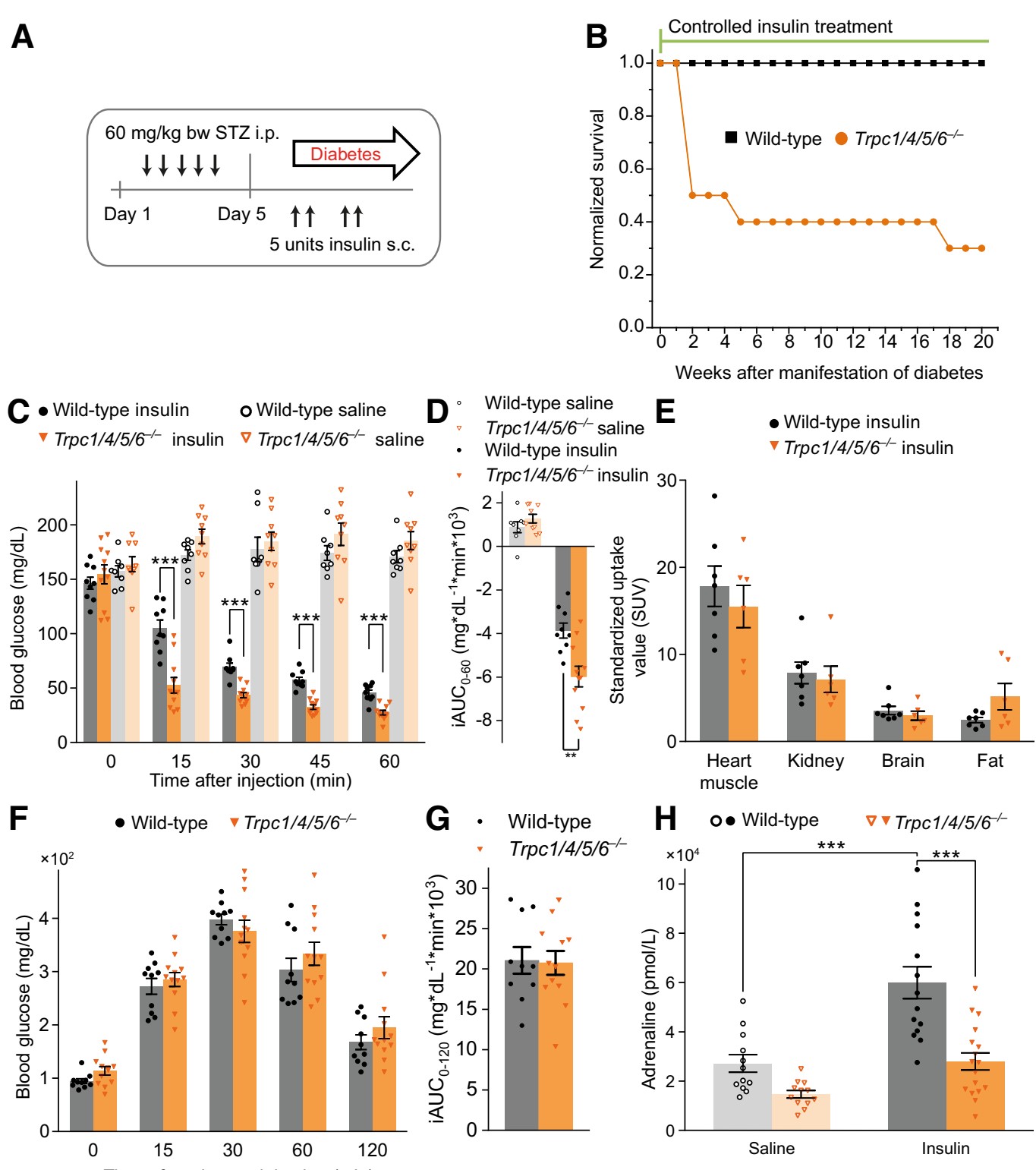

mice lacking TRPC1/4/5 (*Trpc1/4/5*$^{-/-}$), which form hetero-multimeric channel complexes in the murine brain (Bröker-Lai et al, 2017). Since TRPC5 channels have been implicated in glucose metabolism and energy consumption (Gao et al, 2017), we focused with our experiments on the analysis of the *Trpc5* single KO mouse line. Indeed, the genetic ablation of TRPC5 (*Trpc5*$^{-/0}$) sufficed to produce a profound decline in blood glucose levels (Fig. 2A,B). In the same line, the *Trpc5*$^{-/0}$ mice showed significantly lower plasma adrenaline levels compared with the wild-type mice under insulin treatment (Fig. 2C). Moreover, similar results were obtained when

**Figure 1.    Deletion of TRPC channels aggravates insulin-induced hypoglycemia and impairs the associated plasma adrenaline rise.**

(A) Schematic representation of the experimental procedure: Diabetes is induced by i.p injections of streptozotocin (STZ) on five consecutive days. The resultant diabetic hyperglycemia is controlled by s.c. injections of insulin glargin (Lantus®), according to the current blood glucose levels. (B) Survival rate of STZ-induced diabetic mice during the long-term regulation of the blood glucose levels with insulin ($t = 0$: $n = 10$ for $Trpc1/4/5/6^{-/-}$; $n = 10$ for wild-type). After 2 weeks, the insulin dose was lowered for the $Trpc1/4/5/6^{-/-}$ animals (for details see Methods). (C) Insulin tolerance test (ITT): Time course of the blood glucose levels after the i.p. injection of insulin or saline in non-diabetic wild-type or $Trpc1/4/5/6^{-/-}$ mice. Wild-Type vs. $Trpc1/4/5/6^{-/-}$ during insulin treatment at 15 min, $p = 6.55 \times 10^{-5}$; at 30 min $p = 6.93 \times 10^{-6}$; at 45 min $p = 6.08 \times 10^{-7}$; at 60 min $p = 3.72 \times 10^{-5}$. (D) The incremental integrated area under the curve from the ITT in C ($Trpc1/4/5/6^{-/-}$: $n = 11$ for insulin, $n = 9$ for saline; wild-type: $n = 9$ for insulin, $n = 8$ for saline; $p = 0.00286$ for wild-type vs. $Trpc1/4/5/6^{-/-}$ under insulin treatment). (E) Analysis of standardized uptake volumes, using $^{18}$F-fluorodeoxyglucose ($^{18}$F-FDG)-PET. Glucose metabolism rates were unchanged in heart muscle, kidney, brain, and fat tissue of $Trpc1/4/5/6^{-/-}$ ($n = 6$ for $Trpc1/4/5/6^{-/-}$, $n = 7$ for wild-type). (F) Intraperitoneal glucose tolerance test on $Trpc1/4/5/6^{-/-}$ and wild-type animals and (G) incremental area under the curve from the glucose tolerance test ($n = 12$ for $Trpc1/4/5/6^{-/-}$, $n = 10$ for wild-type). (H) Plasma adrenaline levels 60 min after insulin injection for wild-type and $Trpc1/4/5/6^{-/-}$ mice. The plasma levels of mice which received saline intraperitoneally serve as a reference ($Trpc1/4/5/6^{-/-}$: $n = 16$ for insulin, $n = 12$ for saline; wild-type: $n = 14$ for insulin, $n = 12$ for saline; wild-type, insulin vs. saline: $p = 0.000289$; insulin, wild-type vs. $Trpc1/4/5/6^{-/-}$: $p = 3.90 \times 10^{-5}$). **$p < 0.01$, ***$p < 0.001$, two sample $t$ test. Mean ± s.e.m. (C–H) All indicated $n$-values are biological replicates. Source data are available online for this figure.

we compared litter-matched $Trpc5^{-/0}$ and $Trpc5^{+/0}$ males obtained from matings of $Trpc5^{+/-}$ females and $Trpc5^{-/0}$ males, indicating that the observed abnormalities are specifically attributed due to the loss of TRPC5 proteins rather than caused by other potential genetic differences between the mouse lines (Appendix Fig. S1C). Interestingly, Gao et al reported an unaltered insulin-induced decline of the blood glucose levels for their global $Trpc5$ knockout mouse line in which exon 5 of the Trpc5 gene was deleted in mice with a 129/SvImJ background (Groener et al, 2019; Riccio et al, 2009). By contrast, the $Trpc5$ KO we used had a defined C57BL6/N background and had been generated by the excision of exon 4. Therefore, we tested the hypoglycemia phenotype in an independent $Trpc5^{-/0}$ mouse line ($Trpc5^{-/0}$_LB, mixed 129EvSv/C57Bl/6J background), where $Trpc5$ was inactivated by deleting exon 5. The $Trpc5^{-/0}$_LB mice also displayed significantly lower blood glucose levels and a lack of adrenaline elevation compared with the litter-matched $Trpc5^{+/0}$_LB controls (Appendix Fig. S1D,E) (Seemann et al, 2017).

To investigate further the temporal profile of insulin-induced hypoglycemia of TRPC5 mutants and wild-type controls, we extended the observation period of the ITT from 60 to 150 min (Fig. 2A). For wild-type mice, the blood glucose levels dropped for 60 min following insulin injection, before they gradually recovered. In contrast, the blood glucose concentration of $Trpc5^{-/0}$ animals was not only significantly lower in the time period between 15 and 60 min, but further declined to even lower values in the extended time period. Thus, TRPC5-deficient animals display not only more pronounced but also more prolonged hypoglycemia. In close correlation with our preceding experiments, plasma adrenaline levels were profoundly elevated in response to insulin injection in wild-type animals, but not in TRPC5-deficient animals (Fig. 2C). In summary, the results show that TRPC5 channels play a central role in the regulation of hypoglycemia as well as in the reactive increase in plasma adrenaline concentration.

## Adrenaline treatment mitigates aggravated hypoglycemia in Trpc5 KO mice

To test whether the lack of adrenaline rise in TRPC5-deficient animals causes the more severe insulin-induced hypoglycemia, we aimed to prevent aggravated hypoglycemia by supplementing adrenaline shortly after the insulin application in an adrenaline rescue experiment (Fig. 2D). In line with this hypothesis, differences in the glucose values that occurred between mutant

and wild-type mice with insulin administration alone were counterbalanced by adrenaline supplementation for about 30 min. As adrenaline has a very short half-life of 1–2 min (Lip and Hall, 2007), the glucose-rising effect of adrenaline faded after 45 min, showing similarly reduced blood glucose levels in the saline- and in the adrenaline-treated $Trpc5^{-/0}$ group when compared to the corresponding wild-type groups (Fig. 2D). Taken together, these observations highlight a causal link between the absent adrenaline rise and the aggravated hypoglycemia and furthermore show that the catecholamine-evoked blood glucose rise itself is not impaired by $Trpc5$ deletion.

## Pharmacological approaches corroborate the essential role of TRPC5 for the adrenaline homeostasis

To further evaluate the dependence of adrenaline homeostasis on TRPC5 activity, we comparatively analyzed in wild-type and $Trpc5^{-/0}$ animals the functional impact of Englerin A (EA), which directly activates TRPC4/TRPC5 channels (Carson et al, 2015; Cheung et al, 2018). Intraperitoneal injections of EA resulted in wild-type mice after 5 min in a nearly four-fold increase of plasma adrenaline levels compared with the vehicle-treated wild-type group (Fig. 2E). In stark contrast, EA injections failed to induce an increase in plasma adrenaline in TRPC5-deficient animals. Thus, the EA-induced adrenaline elevation relies specifically on the activation of TRPC5 channels. Furthermore, pre-administration of the TRPC5/TRPC4 channel antagonist C31 (Rubaiy et al, 2017) in wild-type mice, caused significantly lower plasma adrenaline 30 min after the insulin application when compared with the corresponding vehicle group (Fig. 2F). Taken together, pharmacological inhibition or genetic deletion of $Trpc5$ leads to a significant reduction in the hypoglycemia-evoked plasma adrenaline rise, corroborating the concept that TRPC5 channel activity controls the adrenaline-mediated counter-regulation in response to hypoglycemia.

## Other stress hormones are not responsible for aggravated hypoglycemia in TRPC-deficient mice

In addition to the HSA axis, hypoglycemic stress activates the hypothalamo-pituitary-adrenocortical (HPA) axis. Accordingly, glucocorticoids and glucagon are also involved in the restoration of euglycemic conditions by increasing the hepatic output of glucose. Still, the plasma corticosterone levels after the injection of

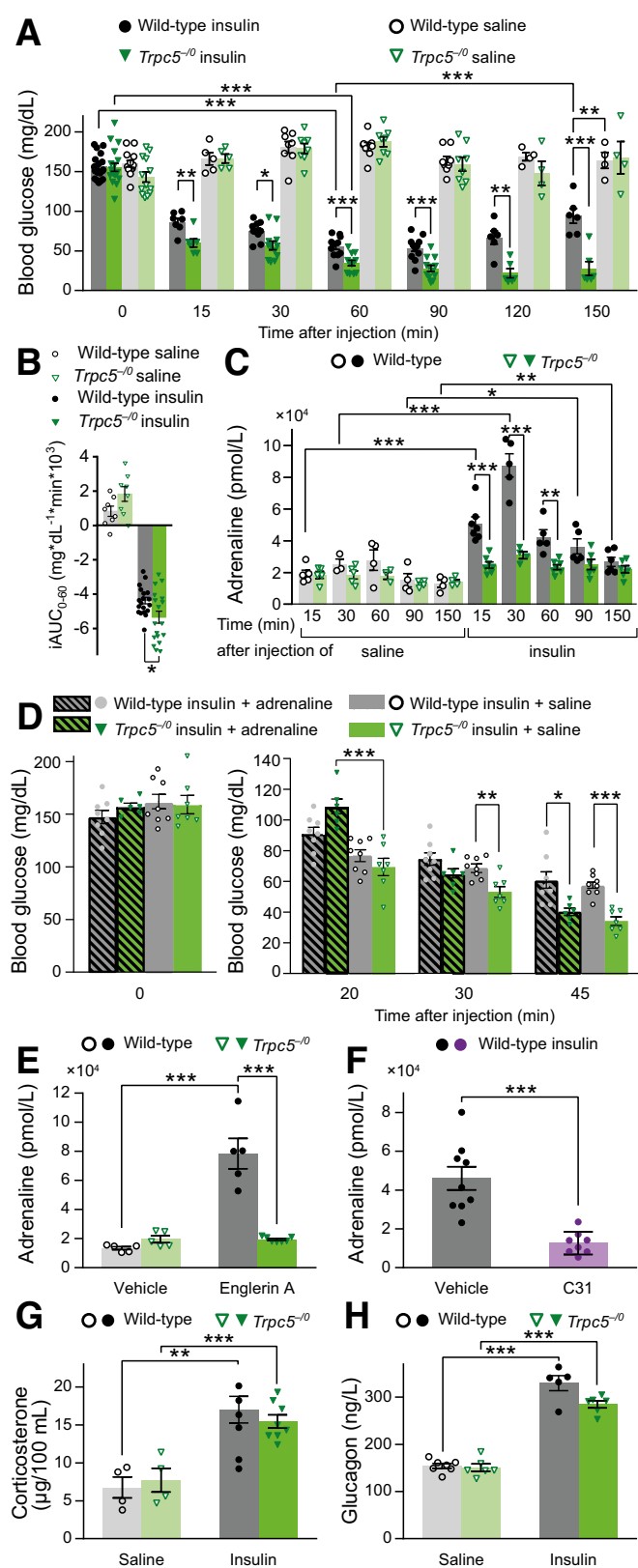

**A** Long-term Insulin tolerance test (ITT). Blood glucose levels of $Trpc5^{-/0}$ mice, measured over 150 min after insulin injection ($Trpc5^{-/0}$ insulin: $n = 19$ (0 min), $n = 7$ (15 min) $n = 12$ (30, 60, 90 min), $n = 6$ (120, 150 min); saline: $n = 19$ (0 min), $n = 7$ (15 min), $n = 8$ (30, 60, 90 min), $n = 4$ (120, 150 min). wild-type insulin: $n = 18$ (0 min), $n = 7$ (15 min), $n = 11$ (30, 60, 90 min), $n = 6$ (120, 150 min); Saline: $n = 13$ (0 min), $n = 5$ (15 min), $n = 8$ (30, 60, 90 min), $n = 4$ (120, 150 min). Wild-type insulin: 0 vs. 15 min $p = 3.31 \times 10^{-10}$, 0 vs. 30 min $p = 1.45 \times 10^{-14}$, 0 vs. 60 min $p = 2.07 \times 10^{-16}$, 0 vs. 90 min $p = 3.36 \times 10^{-16}$, 0 vs. 120 min $p = 1.28 \times 10^{-10}$, 0 vs. 150 min $p = 1.85 \times 10^{-7}$, 60 vs. 120 min $p = 0.000487$, 60 vs. 150 min $p = 1.70 \times 10^{-14}$. $Trpc5^{-/0}$ insulin: 0 vs. 15 min $p = 1.75 \times 10^{-10}$, 0 vs. 30 min $p = 2.13 \times 10^{-13}$, 0 vs. 60 min $p = 8.67 \times 10^{-17}$, 0 vs. 90 min $p = 3.99 \times 10^{-17}$, 0 vs. 120 min $p = 1.48 \times 10^{-12}$, 0 vs. 150 min $p = 1.37 \times 10^{-11}$. Insulin, wild-type vs. $Trpc5^{-/0}$: 15 min $p = 0.00253$, 30 min $p = 0.0102$, 60 min $p = 0.000533$, 90 min $p = 0.000277$, 120 min $p = 0.00110$, 150 min $p = 0.000438$. Wild-type, insulin vs. saline 150 min $p = 0.00116$. **(B)** Incremental integrated area under the curve during a 60 min ITT ($Trpc5^{-/0}$: $n = 18$ for insulin and $n = 8$ for saline; Wild-type: $n = 19$ for insulin and $n = 8$ for saline; wild-type vs. $Trpc5^{-/0}$ with insulin $p = 0.018$). **(C)** Time course of the plasma adrenaline levels corresponding to the ITT in (**A**). ($Trpc5^{-/0}$: for insulin (15 min): $n = 7$; (30 min): $n = 4$; (60 min): $n = 7$; (90, 150 min): $n = 6$; for saline (15, 30 min): $n = 5$; (60 min): $n = 3$; (90, 150 min): $n = 4$; wild-type: for insulin (15 min): $n = 7$; (30, 60, 90 min): $n = 5$; (150 min): $n = 6$; for saline (15 min): $n = 5$; (30 min): $n = 3$; (60, 90, 150 min): $n = 4$). Wild-type, insulin vs. saline: 15 min $p = 0.000187$, 30 min $p = 0.000830$, 90 min $p = 0.0135$, 150 min $p = 0.00848$. Insulin, wild-type vs. $Trpc5^{-/0}$: 15 min $p = 0.000128$, 30 min $p = 0.000303$, 60 min $p = 0.00119$. **(D)** ITT with and without adrenaline supplementation on $Trpc5^{-/0}$ mice. Blood glucose levels of mice receiving injections of insulin plus adrenaline were compared with the ones of mice which received insulin plus saline ($Trpc5^{-/0}$: $n = 6$ for insulin + adrenaline, $n = 7$ for insulin + saline; wild-type: $n = 8$ for insulin + adrenaline, $n = 8$ for insulin + saline). 20 min $Trpc5^{-/0}$, adrenaline vs. saline $p = 0.000459$; 30 min saline, $Trpc5^{-/0}$ vs. wild-type $p = 0.00238$; 45 min $Trpc5^{-/0}$ vs. wild-type: saline $p = 2.87 \times 10^{-5}$, adrenaline $p = 0.0130$. **(E)** Evaluation of the effect of the TRPC5 agonist Englerin A (EA) on plasma adrenaline. Adrenaline levels 5 min after the application of EA or vehicle are shown. EA, Wild-type vs. $Trpc5^{-/0}$ $p = 4.42 \times 10^{-5}$; Wild-type, Vehicle vs. EA $p = 0.000249$ ($Trpc5^{-/0}$: $n = 7$ for EA, $n = 5$ for vehicle; wild-type: $n = 5$ per group). **(F)** Effect of the TRPC5 antagonist C31 on plasma adrenaline levels 30 min after the application of insulin on wild-type mice. The animals received C31 or vehicle prior to insulin ($n = 8$ for C31 + insulin, $n = 9$ for vehicle + insulin, $p = 0.000160$). **(G)** Corticosterone plasma levels 60 min after the injection of insulin or saline ($Trpc5^{-/0}$: $p = 0.000855$, $n = 8$ for insulin, $n = 4$ for saline; wild-type: $p = 0.00309$, $n = 6$ for insulin, $n = 4$ for saline). **(H)** Plasma glucagon levels 60 min after the injection of insulin or saline (Two-way ANOVA, $p = 3.142 \times 10^{-8}$; genotype n.s. $p = 0.316$, treatment $p = 3.20 \times 10^{-11}$; Bonferroni pairwise comparison: $Trpc5^{-/0}$, $p = 1.06 \times 10^{-6}$, $n = 6$ for insulin, $n = 6$ for saline; wild-type: $p = 4.85 \times 10^{-9}$, $n = 5$ for insulin, $n = 7$ for saline). **(A–H)** Mean ± s.e.m., two sample $t$ test, $*p < 0.05$, $**p < 0.01$, $***p < 0.001$ all indicated $n$-values are biological replicates. Source data are available online for this figure.

deletion of *Trpc* genes interferes with the glucocorticoids' action on glucose homeostasis, we subjected the animals to a cortisol tolerance test. Measurements over a two-hour-period revealed that cortisol did not change the blood glucose levels in wild-type or in $Trpc1/4/5/6^{-/-}$ mice compared with the respective saline-treated groups (Appendix Fig. S3E).

The secretion of glucagon can either be stimulated by catecholamines via the HSA axis, by glucocorticoids via the HPA axis, or directly by the hypothalamus. However, the plasma glucagon levels were similarly elevated compared with the saline injections in both the TRPC5-deficient and wild-type mice (Fig. 2H). From these observations, we conclude that the plasma glucagon is not causally associated with the more pronounced blood glucose decline under insulin excess in animals lacking

insulin were similarly elevated in the wild-type and $Trpc5^{-/0}$ mice (Fig. 2G). Thus, aggravated insulin-induced hypoglycemia in *Trpc5* KO animals is not paralleled by up- or downregulated secretion of corticosterone in the plasma. To answer the question whether the

functional TRPC5 channels. We next examined whether the effect of glucagon on glucose homeostasis was impaired in *Trpc5* KO mice. However, the temporary rise of the glucose levels evoked by glucagon injection in both genotypes (at 15 min, Appendix Fig. S3F) illustrates an unchanged glucagon response.

Taken together, neither lower plasma concentrations of glucocorticoids or glucagon nor a reduced systemic response to these hormones are responsible for the phenotype of aggravated insulin-induced hypoglycemia in TRPC5-deficient mice.

### *Trpc5* inactivation in catecholaminergic cells aggravates hypoglycemia and abolishes adrenaline elevation

We next questioned whether the processes of hypoglycemia counter regulation may be affected by *Trpc* inactivation in the central nervous system (Fig. 3A). First, we examined the potential role of TRPC5 channels in cholinergic splanchnic nerves (2 in Fig. 3A) by crossing the ChAT-Cre line (Lowell et al, 2006; Rossi et al, 2011) with *Trpc5fx/0* mice to generate animals in which *Trpc5* was selectively inactivated in the ACh-producing nerve cells (*Trpc5fx/0;ChAT-Cre+*). Yet, blood glucose and adrenaline levels of the *Trpc5fx/0;ChAT-Cre+* mice were not lower than those of the *Trpc5fx/0;ChAT-Cre-* controls (Fig. 3B–D). Thus, the deletion of *Trpc5* in cholinergic neurons does not result into the phenotype observed upon ubiquitous *Trpc5* inactivation.

To address the question whether TRPC5 in catecholaminergic cell types (3, 4 in Fig. 3A) are involved in the counter-regulatory circuits, we used mice that express Cre under control of the dopamine-β-hydroxylase (DBH) promotor (Parlato et al, 2007), which is active in all adrenaline- and noradrenaline-producing neurons in the CNS, as well as in neuroendocrine chromaffin cells in the adrenal medulla (Lewis et al, 1990). In response to an ITT, the phenotype of *Trpc5fx/0;DBH-Cre+* animals closely resembled the one of the global *Trpc5* KO (*Trpc5-/0*). Compared with the *Trpc5fx/0;DBH-Cre-* mice, the *Trpc5fx/0; DBH-Cre+* mice exhibited more severe hypoglycemia paralleled by significantly lower plasma adrenaline levels (Fig. 4A–F). These effects were observed in both male (Fig. 4A–C) and female (Fig. 4D–F) mice, showing no difference between sexes (two-way ANOVA on the iAUC $p = 0.074$ (n.s.) for sex, $p = 1.5 \times 10^{-5}$ (***) for genotype). Furthermore, in hyperinsulinemic hypoglycemic clamp experiments the glucose infusion rate was significantly higher in *Trpc5fx/0;DBH-Cre+* compared to *Trpc5fx/0;DBH-Cre-* mice to achieve isoglycemia at 50 mg/dL blood glucose (Appendix Fig. S3A,B), indicating the inability of *Trpc5fx/0;DBH-Cre+* mice to autonomously increase their plasma glucose levels under hypoglycemic conditions.

Taken together, these findings assign an essential role to TRPC5 for the counter regulation in catecholaminergic cells - most likely in RVLM neurons of the hindbrain or in chromaffin cells of the adrenal medulla.

### Genetic loss of *Trpc5* in RVLM neurons does not impair autonomous counter regulation

We next evaluated a potential role of TRPC5 in catecholaminergic RVLM neurons in hypoglycemia counter-regulation (Ritter et al, 2011; Verberne et al, 2014; Zhao et al, 2017). For this, we injected Adeno-associated viruses (AAV2) expressing a Cre-GFP fusion protein in the ventrolateral medulla of *Trpc5fx/0* mice (Appendix Fig. S4A,B). The transduction rate of catecholaminergic neurons was assessed three weeks after the stereotactic injections by the number of tyrosine hydroxylase (TH)-sensitive neurons (in red)

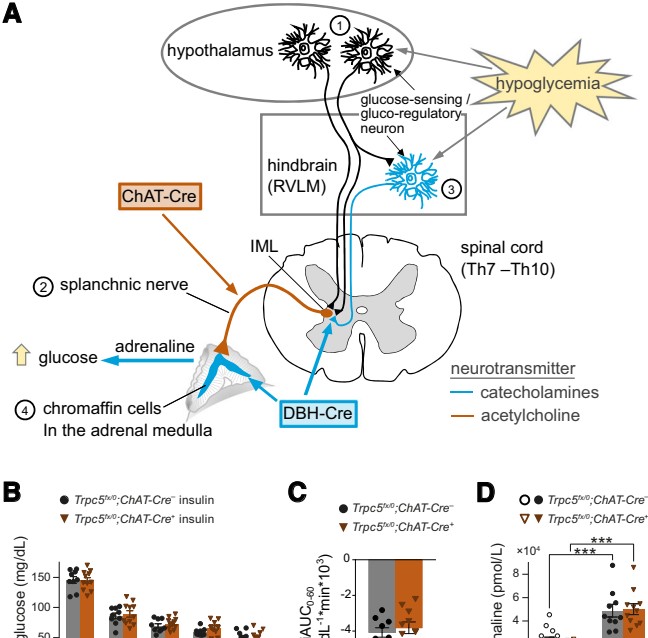

**Figure 3. Tissue-specific *Trpc5*-deletion and its effect on autonomic counter regulation.**

(A) Schematic representation of the processes underlying the central stress perception and stress response to hypoglycemia. IML intermediolateral nucleus, Th thoracic vertebrae, RVLM ventrolateral medulla. (B) Time course of blood glucose levels during an insulin tolerance test in a mouse line with *Trpc5* deletion in cholinergic neurons (male *Trpc5fx/0;ChAT-Cre+: n = 11; Trpc5fx/0;ChAT-Cre-: n = 9*). (C) Incremental integrated area under the curve during the ITT. (D) Plasma adrenaline levels 30 min after insulin injection (male *Trpc5fx/0;ChAT-Cre+: p = 8.19 × 10^{-7}, n = 11* for insulin, *n = 11* for saline; *Trpc5fx/0;ChAT-Cre-: p = 0.000217, n = 10* for insulin, *n = 13* for saline). ***$p < 0.001$, two sample t-test. (B–D) Mean ± s.e.m. all indicated *n*-values are biological replicates. Source data are available online for this figure.

that were also positive for GFP (in green) in relation to all TH-neurons in the RVLM (Appendix Fig. S4B, 59 ± 3%). The *Trpc5fx/0* control group received AAVs expressing GFP without Cre. Three weeks after the AAV treatment, blood glucose levels in response to insulin application did not differ between the AAV-Cre-GFP injected *Trpc5fx/0* mice and the AAV-GFP injected controls (Appendix Fig. S4C). Similarly, 30 min after insulin injection, the plasma adrenaline levels remained unchanged (Appendix Fig. S4D). Immunohistological analyses of TRPC5 expression in the VLM revealed that only few TRPC5-expressing cells were TH-positive (Appendix Fig. S4E). Furthermore, no significant co-localization of TRPC5 and TH was found in the in the nucleus tractus solitarii (NTS) (Appendix Fig. S4F), arguing against a functional role of TRPC5 in these catecholaminergic neurons.

### Loss of TRPC5 activity in chromaffin cells strongly impairs PACAP- and muscarine-evoked catecholamine secretion

To test whether TRPC5 may control the autonomous counter-regulation and restoration of euglycemia by modulating adrenaline

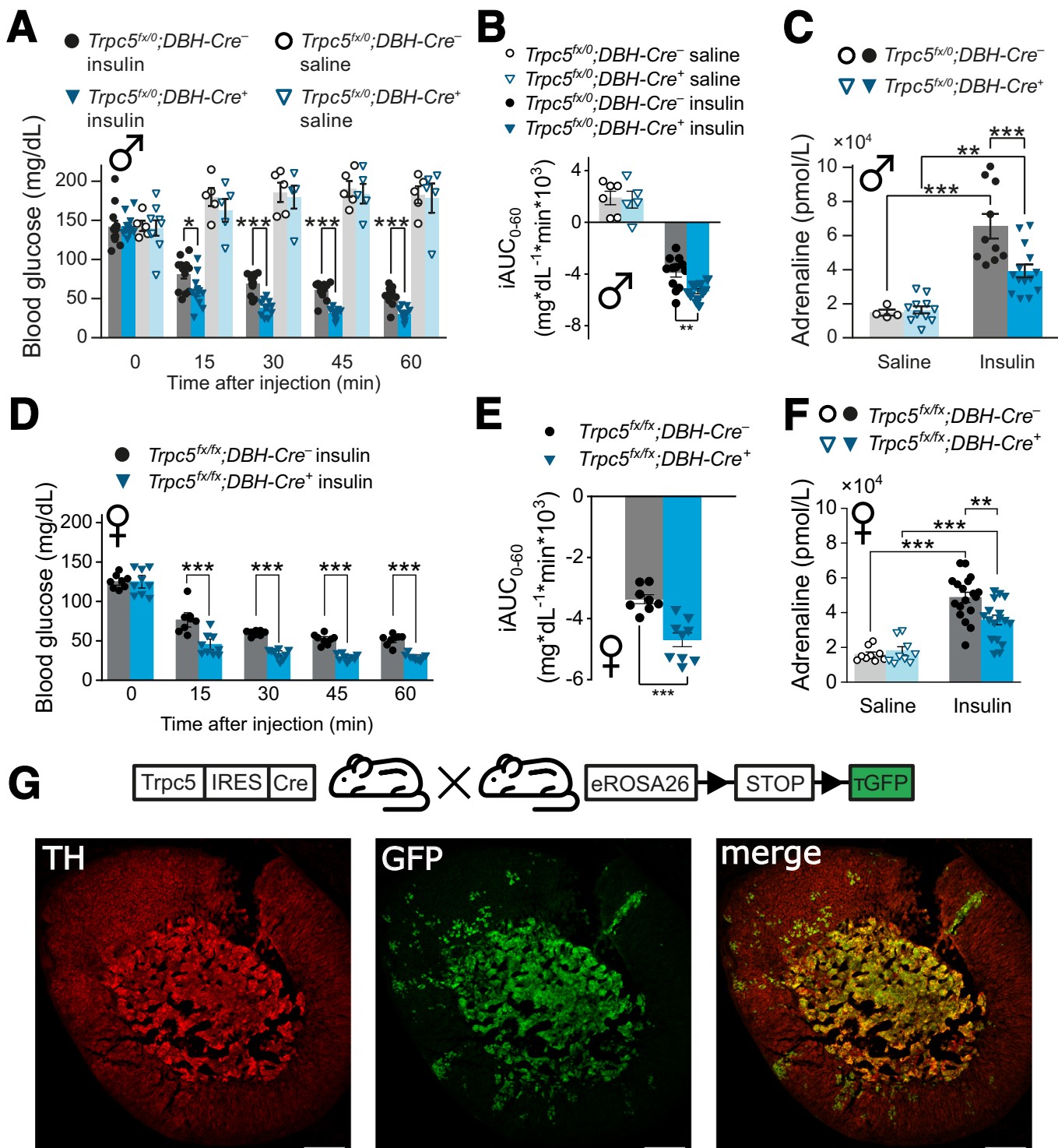

secretion from chromaffin cells of the adrenal medulla, we made use of the recently generated TRPC5 reporter mouse, where Cre recombinase is expressed under the control of a *Trpc5* promoter (Wyatt et al, 2017). TRPC5-positive cells were visualized by detection of Cre-dependent GFP reporter fluorescence (Fig. 4G) in animals that express Cre recombinase under the control of the *Trpc5* promotor (*Trpc5-IRES-Cre; eR26-τGFP*). In the medulla,

virtually all cells exhibited GFP fluorescence (Fig. 4G, middle) and were TH-positive (Fig. 4G, left), as evident from the merged images (Fig. 4G, right). The splanchnic-adrenal synapse co-releases ACh and the neuropeptide PACAP (Borges et al, 2018; Eiden et al, 2018), which may engage TRPC5 channels via M1 or PAC1 receptors in chromaffin cells. To test this hypothesis, we comparatively analyzed $Ca^{2+}$ signals in response to muscarine or

**Figure 4.  TRPC5 channels in catecholaminergic cells are relevant for autonomic counter regulation.**

(A) Time course of blood glucose levels during insulin tolerance tests in a mouse line with $Trpc5$ deleted in catecholaminergic cells. Male $Trpc5^{fx/0};DBH\text{-}Cre^+$: $n = 11$ for insulin, $n = 5$ for saline; $Trpc5^{fx/0};DBH\text{-}Cre^-$: $n = 12$ for insulin, $n = 6$ for saline. $Trpc5^{fx/0};DBH\text{-}Cre^+$ vs. $Trpc5^{fx/0};DBH\text{-}Cre^-$ after insulin injection, $p = 0.0194$ (15 min), $p = 3.12 \times 10^{-6}$ (30 min), $p = 1.05 \times 10^{-7}$ (45 min), $p = 1.11 \times 10^{-5}$ (60 min). (B) The incremental integrated area under the curve from (A). Male $Trpc5^{fx/0};DBH\text{-}Cre^+$: $n = 11$ for insulin, $n = 5$ for saline; $Trpc5^{fx/0};DBH\text{-}Cre^-$: $n = 12$ for insulin, $n = 6$ for saline. $Trpc5^{fx/0};DBH\text{-}Cre^+$ vs. $Trpc5^{fx/0};DBH\text{-}Cre^-$ after insulin injection, $p = 0.00219$. (C) The plasma adrenaline levels 30 min after insulin injection. Male $Trpc5^{fx/0};DBH\text{-}Cre^+$: $p = 6.72 \times 10^{-5}$, $n = 14$ for insulin, $n = 11$ for saline; $Trpc5^{fx/0};DBH\text{-}Cre^-$: $p = 0.000966$, $n = 10$ for insulin, $n = 4$ for saline; insulin treatment between genotypes $p = 0.00205$. (D) As (A) but in female mice. Female $Trpc5^{fx/fx};DBH\text{-}Cre^+$: $n = 9$; $Trpc5^{fx/fx};DBH\text{-}Cre^-$: $n = 8$, $p = 0.000792$ (15 min), $p = 2.50 \times 10^{-9}$ (30 min), $p = 4.20 \times 10^{-7}$ (45 min), $p = 9.11 \times 10^{-7}$ (60 min). (E) The incremental integrated area under the curve from (D). Female $Trpc5^{fx/fx};DBH\text{-}Cre^+$: $n = 9$; $Trpc5^{fx/fx};DBH\text{-}Cre^-$: $n = 8$, $p = 0.000226$. (F) The plasma adrenaline levels 30 min after insulin injection for female $Trpc5^{fx/fx};DBH\text{-}Cre^+$: $p = 0.000310$, $n = 17$ (insulin), $n = 9$ (saline) and $Trpc5^{fx/fx};DBH\text{-}Cre^-$: $p = 5.49 \times 10^{-8}$, $n = 18$ (insulin), $n = 9$ (saline). Insulin treatment between genotypes: $p = 0.0030$. (G) Representative immunohistology of adrenal gland sections, using anti-TH antibodies (red) and anti-GFP (green) in $Trpc5\text{-}IRES\text{-}Cre;eR26\text{-}\tau GFP$ animals where GFP serves as an indicator for TRPC5 expression. Scale bar = 200 μm. (A–F) Mean ± s.e.m., $*p < 0.05$, $**p < 0.01$, $***p < 0.001$, two sample $t$ test, all indicated $n$-values are biological replicates. Source data are available online for this figure.

PACAP in acutely isolated chromaffin cells from the $Trpc1/4/5^{-/-}$, $Trpc5^{-/0}$ and wild-type mice using Fura-2-based fluorescence measurements. Application of either agonist evoked a strong rise in the intracellular $Ca^{2+}$ concentration ($[Ca^{2+}]_i$), which was greatly diminished in $Trpc1/4/5^{-/-}$ or $Trpc5^{-/0}$ chromaffin cells (Fig. 5A–D). Furthermore, direct activation of TRPC5 channels by Englerin A (EA) similarly increased $[Ca^{2+}]_i$, an effect that was fully abolished in $Trpc1/4/5^{-/-}$ or $Trpc5^{-/0}$ deficient cells (Fig. 5E,F). In contrast, stimulation of nicotinergic receptors in chromaffin cells led to an unchanged rise of $[Ca^{2+}]_i$ in all groups, demonstrating that TRPC5 deficiency has a profound impact on cytosolic $Ca^{2+}$ dynamics by specific interference with metabotropic, but not with ionotropic signaling at the splanchnic-adrenomedullary synapse (Fig. 5G,H).

TRPC5-deficient animals exhibit striking parallels to PACAP-deficient mice in terms of recovery from insulin-induced hypoglycemia (Hamelink et al, 2002; Stroth et al, 2011). To further explore the role of TRPC5 channels, we analyzed whether PACAP-induced exocytosis is altered in chromaffin cells from TRPC5-deficient mice using carbon fiber amperometry. Wild-type cells strongly responded to PACAP superfusion with a barrage of exocytotic events, reflecting catecholamine secretion from chromaffin granules (Fig. 5I). In contrast, the number of PACAP-evoked events was substantially reduced in $Trpc5^{-/0}$ cells, indicating that the ligand-mediated activation of TRPC5 channels is responsible for sustained catecholamine secretion in wild-type cells (Fig. 5I–K). Neither the amplitude nor the kinetics of the individual release events was changed in the $Trpc5$ KO cells, demonstrating that the properties of the single secretion events remain unaffected by TRPC5 activity (Appendix Fig. S5A–D). In the same line, the pharmacological inhibition of TRPC5 channels by the antagonist HC-070 (Bauer et al, 2020) phenocopies the strong inhibition observed in the absence of TRPC5. Thus, acute loss of TRPC5 activity in chromaffin cells rather than potential developmental adaptions in the absence of TRPC5 proteins can be held responsible for the observed phenotype. Moreover, similar results were obtained when we comparatively analyzed the secretory properties of cells in which the $Trpc5$ gene was excised by the Cre recombinase under the control of the DBH promotor ($Trpc5^{fx/0};DBH\text{-}Cre^+$) and control cells ($Trpc5^{fx/0};DBH\text{-}Cre^-$, Fig. 5L–N; Appendix Fig. S5E–G). The results corroborate that the specific loss of TRPC5 activity in chromaffin cells, rather than TRPC5-mediated alterations in splanchnic nerve firing, can account for the sharp decrease in catecholamine secretion. Thus, TRPC5 channels are crucial mediators for the PACAP-induced sustained catecholamine release from chromaffin cells. Muscarine has been shown to activate M1

receptors in chromaffin cells, which stimulate an unidentified non-selective ion conductance and drive sustained catecholamine secretion (Calvo-Gallardo et al, 2016; Harada et al, 2015; Inoue et al, 2018). To study a possible engagement of M1 receptors in TRPC5-mediated secretion, we applied muscarine (30 μM), which evoked a barrage of amperometric events, that were abolished in the absence of TRPC5 channels (Fig. 5O–Q). Again, neither the amplitude nor the kinetics of the individual amperometric spikes were changed in the absence of the TRPC5 channels (Appendix Fig. S6). Taken together, signaling pathways of potent secretagogues like muscarine and PACAP engage TRPC5 channels as common effector molecules to drive sustained catecholamine release from chromaffin cells.

To further explore the mode of TRPC channel activation, we directly recorded channel activity in chromaffin cells. For this, we used the perforated patch clamp technique to maintain the integrity of cytoplasmic components including soluble second messengers (Lippiat, 2008). PACAP application evoked burst-like inward currents, which were absent in TRPC1/4/5-deficient cells (Fig. 6A–F). In close correlation with our amperometric recordings, PACAP application caused a robust increase in membrane capacitance in wt cells, indicating granule exocytosis, which was strongly reduced for TRPC1/4/5-deficient cells (Fig. 6G). Moreover, changes in the integrated current response correlated well with increases in membrane capacitance (Fig. 6H). Furthermore, we were able to isolate PACAP-induced TRPC5 currents in chromaffin cells + in the whole-cell configuration (Appendix Fig. S7A,B). In response to a voltage ramp protocol, (in the presence of TTX, TEA and $Cs^+$) clear inward and outward currents were observed in wt cells with a reversal potential of $-11$ mV indicative of a non-selective cation conductance. In cells isolated from $Trcp5^{-/0}$ mice, the non-selective cation conductance was abolished, indicating its dependence on TRPC5 activity. Moreover, the observed PACAP-induced TRPC5 currents agree well with previous published TRPC5-dependent currents (Blair et al, 2009; Riccio et al, 2009). Thus, PACAP stimulates TRPC channel activity, which in turn suffices to trigger secretion in mouse chromaffin cells. Similar results were obtained when we repeated these experiments in chromaffin cells from the $Trpc5$ single KO mouse line. In comparison with wt cells (Fig. 7A), genetic loss of TRPC5 fully abolished any current activation and diminished the capacitance increase in response to PACAP application (Fig. 7B,D–F). Furthermore, treatment with the phospholipase C (PLC) blocker U73122 (10 μM) strongly reduced the PACAP-evoked current activity and the increase in membrane capacitance, indicating that PLC

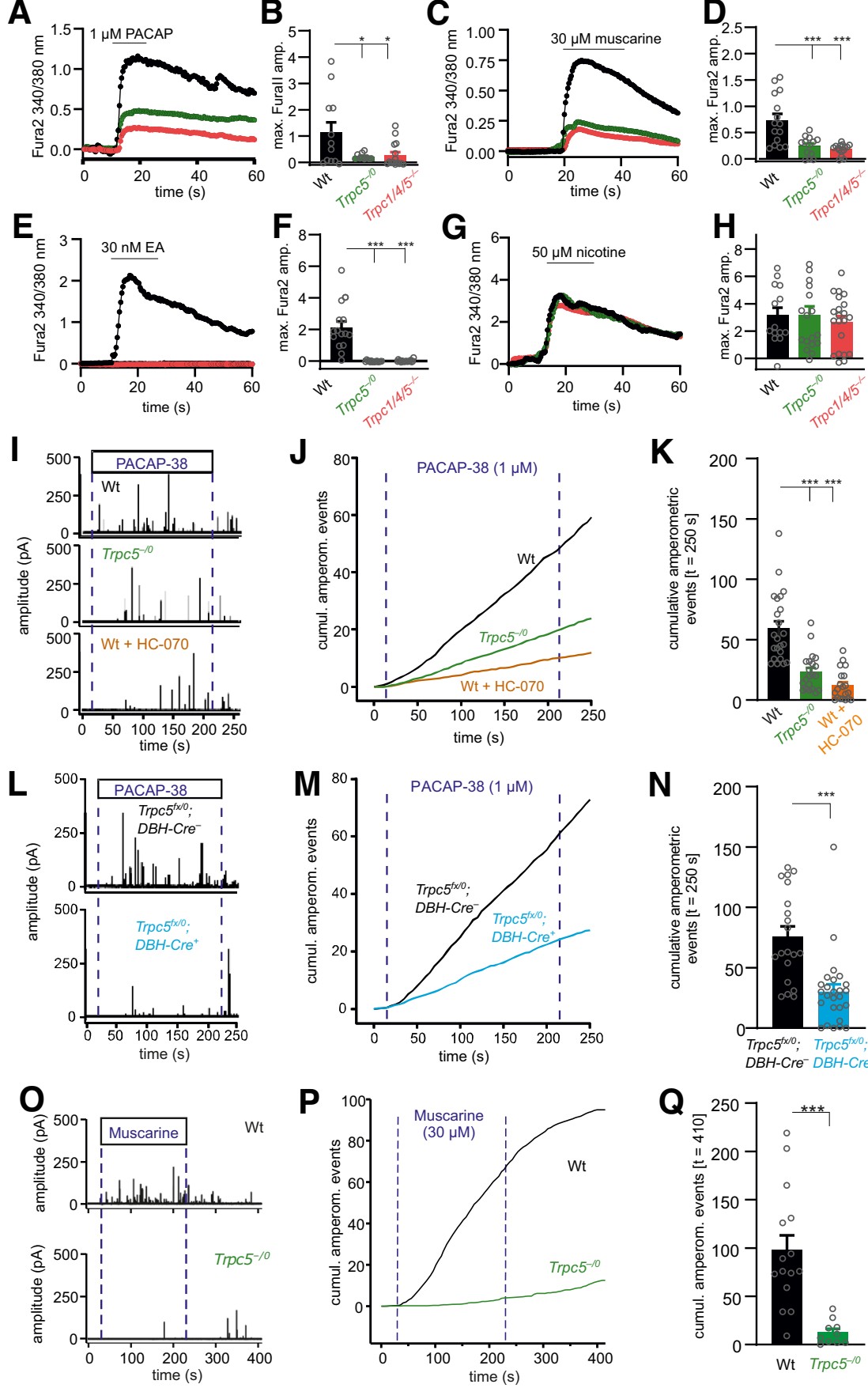

**Figure 5.  Genetic loss or pharmacological inhibition of TRPC5 strongly reduces muscarine and PACAP-evoked catecholamine secretion from chromaffin cells.**

(A) Average time course of the intracellular calcium concentration in Fura-2-AM-loaded chromaffin cells stimulated with 1 µM PACAP. (B) The increase in $[Ca^{2+}]_i$ is significantly reduced in $Trpc1/4/5^{-/-}$ (red, $p = 0.0306$) and TRPC5-deficient ($Trpc5^{-/0}$, green, $p = 0.0395$) cells when cells were perfused with 1 µM PACAP. Wt, $n = 12$; $Trpc5^{-/0}$, $n = 10$; $Trpc1/4/5^{-/-}$, $n = 15$ from 4 independent preparations. (C) As (A) but with 30 µM muscarine. (D) The increase in $[Ca^{2+}]_i$ is significantly reduced in $Trpc1/4/5^{-/-}$ (red, $p = 4.22 \times 10^{-5}$) and TRPC5-deficient ($Trpc5^{-/0}$, green, $p = 3.70 \times 10^{-4}$) cells when cells were perfused with 30 µM muscarine. wt, $n = 15$; $Trpc5^{-/0}$, $n = 13$; $Trpc1/4/5^{-/-}$, $n = 15$ from 3 independent preparations. (E) As (A) but with 30 nM Englerin A (EA). (F) The increase in $[Ca^{2+}]_i$ is significantly reduced in $Trpc1/4/5^{-/-}$ (red, $p < 2 \times 10^{-16}$) and TRPC5-deficient ($Trpc5^{-/0}$, green, $p < 2 \times 10^{-16}$) cells when cells were perfused with 30 nM EA. wt, $n = 15$; $Trpc5^{-/0}$, $n = 15$; $Trpc1/4/5^{-/-}$, $n = 19$ from 3 independent preparations. (G) As (A) but with 50 µM nicotine. (H) The increase in $[Ca^{2+}]_i$ remained unaffected by nicotinergic stimulation. wt, $n = 14$; $Trpc5^{-/0}$, $n = 18$; $Trpc1/4/5^{-/-}$, $n = 21$. One-way ANOVA with Tukey post hoc means comparison, *$p < 0.05$, ***$p < 0.001$ (B, D, F). (I–Q) Amperometric analysis of catecholamine secretion from chromaffin cells isolated from the adrenal medulla. (I) Exemplary amperometric recordings in response to PACAP application for $Trpc5^{-/0}$ and wild-type chromaffin cells and wild-type cells treated with HC-070 (50 nM). Dashed blue lines indicate the beginning and the end of PACAP application (1 µM, 200 s). (J) Cumulative presentation of averaged secretion in response to PACAP. (K) Loss of TRPC5 or its acute inhibition strongly reduces PACAP-evoked secretion. Data were collected from wt, $n = 23$; $Trpc5^{-/0}$, $n = 21$, $p = 6.00 \times 10^{-7}$; wt+HC-070, $n = 20$, $p < 2 \times 10^{-16}$ from 3 independent preparations. One-way ANOVA Tukey Kramer post hoc vs. wt. ***$p < 0.001$. (L) Exemplary amperometric recordings of chromaffin cells prepared $Trpc5^{fx/0};DBH-Cre^-$ and $Trpc5^{fx/0};DBH-Cre^+$: mice in response to PACAP application. (M) Mean cumulative plot of amperometric events for the indicated groups. (N) Total number of amperometric events after 250 s for the indicated groups. Data were collected from $Trpc5^{fx/0};DBH-Cre^-$, $n = 20$; $Trpc5^{fx/0};DBH-Cre^+$, $n = 26$. Mann–Whitney U test, $p = 6.29 \times 10^{-5}$, ***$p < 0.001$. (O) Exemplary amperometric recordings in response to muscarine application for wt and $Trpc5^{-/0}$ cells. (P) Mean cumulative plot of amperometric events for the indicated groups. Dashed blue lines indicate the beginning and the end of muscarine application (30 µM, 200 s). (Q) Total number of amperometric events after 410 s for the indicated groups. Data were collected from wt, $n = 15$; $Trpc5^{-/0}$, $n = 10$ from 2 independent preparations. Mann–Whitney U test, $p = 1.70 \times 10^{-4}$, ***$p < 0.001$. (B, D, F, H, K, N, Q) Mean ± s.e.m. Source data are available online for this figure.

activation mediates an essential step in the PAC1 receptor-TRPC5 channel coupling (Fig. 7C–F). The PLC blocker U73122 also abolished PACAP-evoked catecholamine secretion as judged from our amperometric recordings, suggesting that the reduced increase in membrane capacitance is due to hindered chromaffin granule exocytosis (Appendix Fig. S7C–E).

To investigate how TRPC5 channel activation affects the membrane potential, we studied chromaffin cells in the current-clamp configuration using the perforated-patch clamp condition (Fig. 8). Starting from similar resting potentials for wt ($-71 \pm 1.9$ mV, $n = 14$) and TRPC5 KO cells ($-70 \pm 2.0$ mV, $n = 17$), PACAP stimulation led to a robust depolarization of the membrane potential in wt cells ($\Delta$VM: $22.4 \pm 2.4$ mV), which was almost completely abolished in TRPC5 ko cells ($\Delta$VM: $4.0 \pm 2.0$ mV, Fig. 8A,B). Notably, the PACAP-evoked membrane depolarization was long lasting, but remained sub-threshold (Fig. 8C), consistent with previous observations (Kuri et al, 2009). Thus, TRPC5 activation elicited by PACAP depolarizes chromaffin cells, but does not result in the initiation of action potentials. Overall, these results together with the in vivo experiments delineate a hitherto unrecognized signaling pathway for the observed rise in plasma adrenaline in response to insulin-induced hypoglycemia.

## Metabolic parallels between TRPC5 deficiency and HAAF

The impaired counter-regulatory adrenaline response to declining glucose levels caused by the lack of functional TRPC5 proteins resembles a pathological condition in diabetes patients called HAAF (Cryer, 2005). In HAAF patients, recurrent hypoglycemic episodes cause a readjustment of the threshold below which sympathetic counter-regulation is induced (Senthilkumaran et al, 2016). Therefore, we attempted to reveal commonalities between TRPC5 deficiency in mice and HAAF in diabetes patients. To this end, we compared changes in hypoglycemia-evoked metabolic signatures in the plasma as circulating catecholamine levels influence the hepatic output and thereby changes in metabolite concentrations. Under insulin-induced hypoglycemia, we found that the concentration of numerous amino acids was reduced in $Trpc5^{fx/0};DBH-Cre^+$ mice (Appendix Fig. S8A) when compared with the control group ($Trpc5^{fx/0};DBH-Cre^-$) or with plasma taken from

saline injected $Trpc5^{fx/0};DBH-Cre^+$ mice. Similar reductions were found for the TCA cycle metabolites fumarate and malate (Appendix Fig. S8B). The levels of ketoglutarate and pyruvate, as well as those of the fatty acids palmitoleic, myristic, linoleic, and oleic acid, were comparably lowered by the insulin treatment in both $Trpc5^{fx/0};DBH-Cre^+$ and $Trpc5^{fx/0};DBH-Cre^-$ animals (Appendix Fig. S8C). In the plasma of male and female HAAF patients, the levels of most metabolites were comparable to those of the non-HAAF patients with diabetes before and during controlled hypoglycemia (60–70 mg/dL) (Groener et al, 2019) (Appendix Fig. S9), except for the three amino acids lysine, aspartic acid, and taurine. While lysine plasma concentrations were reduced under baseline conditions in the HAAF patients (Fig. 9A), lysine was only lowered by the induction of hypoglycemia in $Trpc5^{fx/0};DBH-Cre^+$ mice (Fig. 9B). Aspartic acid levels were reduced in the HAAF patients under hypoglycemia compared with the non-HAAF patients with diabetes, but were not changed under any of the conditions measured in the TRPC5-deficient animals. Notably, under hypoglycemia, taurine plasma levels were significantly lowered in the HAAF patients and in the $Trpc5^{fx/0};DBH-Cre^+$ mice. We did not observe sex-dependent differences in the patient population. This analogy suggests that taurine and taurine-dependent metabolic pathways may be commonly impaired in both cases of hypoglycemia-induced autonomic failure.

## Discussion

The autonomous counter regulation evoked by hypoglycemia is an essential physiological process that is impaired in diabetes patients suffering from HAAF. Recently, Zhao et al deciphered that catecholaminergic neurons in the RVLM are crucial in mediating stress-induced hyperglycemia by stimulating the secretion of catecholamines from the adrenal medulla (Zhao et al, 2017). So far, little is known about the molecular entities that determine the release of catecholamines in this counter-regulatory pathway. In the present study, we establish that TRPC channels in chromaffin cells of the adrenal medulla play a pivotal role in the adrenaline-dependent counter regulation to

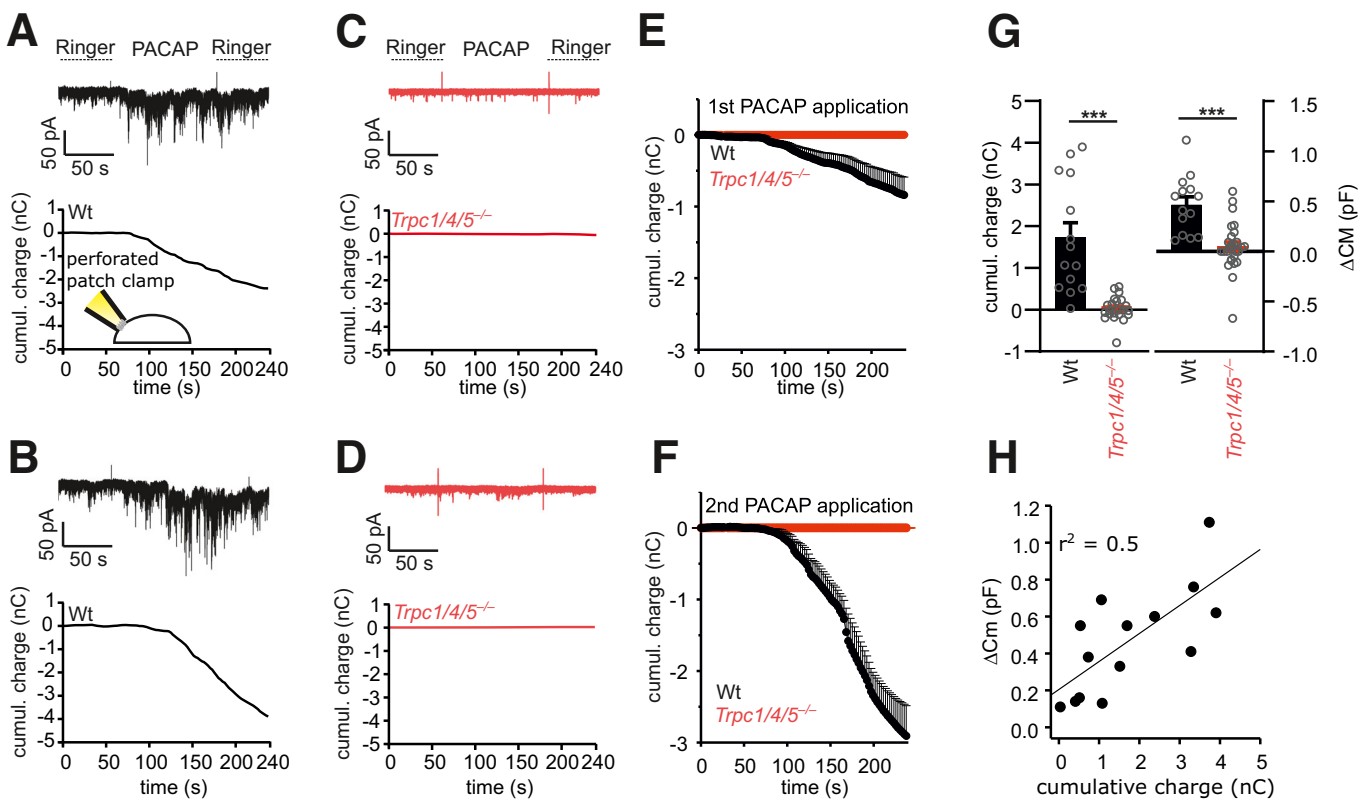

**Figure 6. PACAP evokes TRPC channel mediated inward currents and secretion in mouse chromaffin cells.**

(A) Exemplary recordings (perforated-patch configuration) of a PACAP-evoked inward current at a holding potential of −70 mV in a wt cell (upper panel) and its cumulative charge plot (lower panel). Note the burst-like current deflections. (B) Repeated application of PACAP to the same cell shown in (A). (C, D) No significant conductance changes could be evoked in $Trpc1/4/5^{-/-}$ cells (1st, (C) and 2nd application (D)). (E, F) Time course of mean charge transfer for the 1st (E) and 2nd application of PACAP (F). (G) The PACAP-evoked charge transfer (left panel $p = 6.43 \times 10^{-6}$) and increase in membrane capacitance (right panel $p = 2.75 \times 10^{-5}$) are strongly reduced in $Trpc1/4/5^{-/-}$ cells. (H) Changes in membrane capacitance (indicative of exocytosis) correlate well with the charge transfer evoked by PACAP. (E–H) Data were collected from 12 WT and 15 $Trpc1/4/5^{-/-}$ cells from 3 independent preparations, mean ± s.e.m., ***$p < 0.001$, Mann–Whitney U test. Source data are available online for this figure.

hypoglycemia. First, insulin injection caused increased mortality and a more severe and prolonged decline of blood glucose levels in $Trpc1/4/5/6^{-/-}$ mice. Second, the impaired glucose homeostasis is due to an absent adrenaline elevation in response to insulin-evoked hypoglycemia, as observed for the impaired autonomous failure in HAAF patients (Cryer, 2013). Third, the hypoglycemia phenotype was pinpointed to the functional loss of TRPC5 channel activity specifically in chromaffin cells. Fourth, PACAP and muscarine stimulated sustained secretion of adrenaline from the chromaffin cells critically depends on TRPC5 channel activity. Fifth, our analysis of the metabolic signatures in the $Trpc5^{fx/0};DBH\text{-}Cre^+$ mice and HAAF patients revealed a drop in plasma taurine levels under hypoglycemia as a commonality between both pathological conditions, opening up new diagnostic avenues to identify HAAF risk patients based on altered plasma taurine levels.

## Identifying the relevant site of TRPC action in the adrenaline-mediated counter regulation

As an explanation for the aggravated hypoglycemia and the concomitant impairment in plasma adrenaline rise in the TRPC-deficient mice, a

role for TRPC channels was conceivable at multiple steps in the signaling cascades underlying the autonomous counter regulation. To address the possibility that the more pronounced hypoglycemia in animals lacking TRPCs results from increased peripheral insulin sensitivity, we subjected mice to an oral glucose tolerance test. Here, we observed unchanged glucose clearance. In addition, we used PET scans to analyze insulin-evoked glucose ($^{18}$F-FDG) uptake into organs with high insulin sensitivity or high energy consumption. Comparable results in the $Trpc1/4/5/6^{-/-}$ and wild-type mice also argue against an enhanced insulin-sensitive uptake of glucose into the mutant cells as an explanation for the aggravated hypoglycemia under insulin treatment.

Hypoglycemic stress activates both the HSA and HPA axes. The relevance of alterations in the HPA axis was assessed by investigations of glucocorticoid and glucagon regulation and action. However, neither lower plasma concentrations of glucocorticoids or glucagon in the ITT experiments nor a reduced organismal response to these hormones are responsible for the phenotype of aggravated insulin-induced hypoglycemia observed in the TRPC-deficient mice. To corroborate this conclusion we also measured glucagon and corticosterone level during hypoglycemic clamp experiments when glucose levels were identical at 50 mg/dL in both

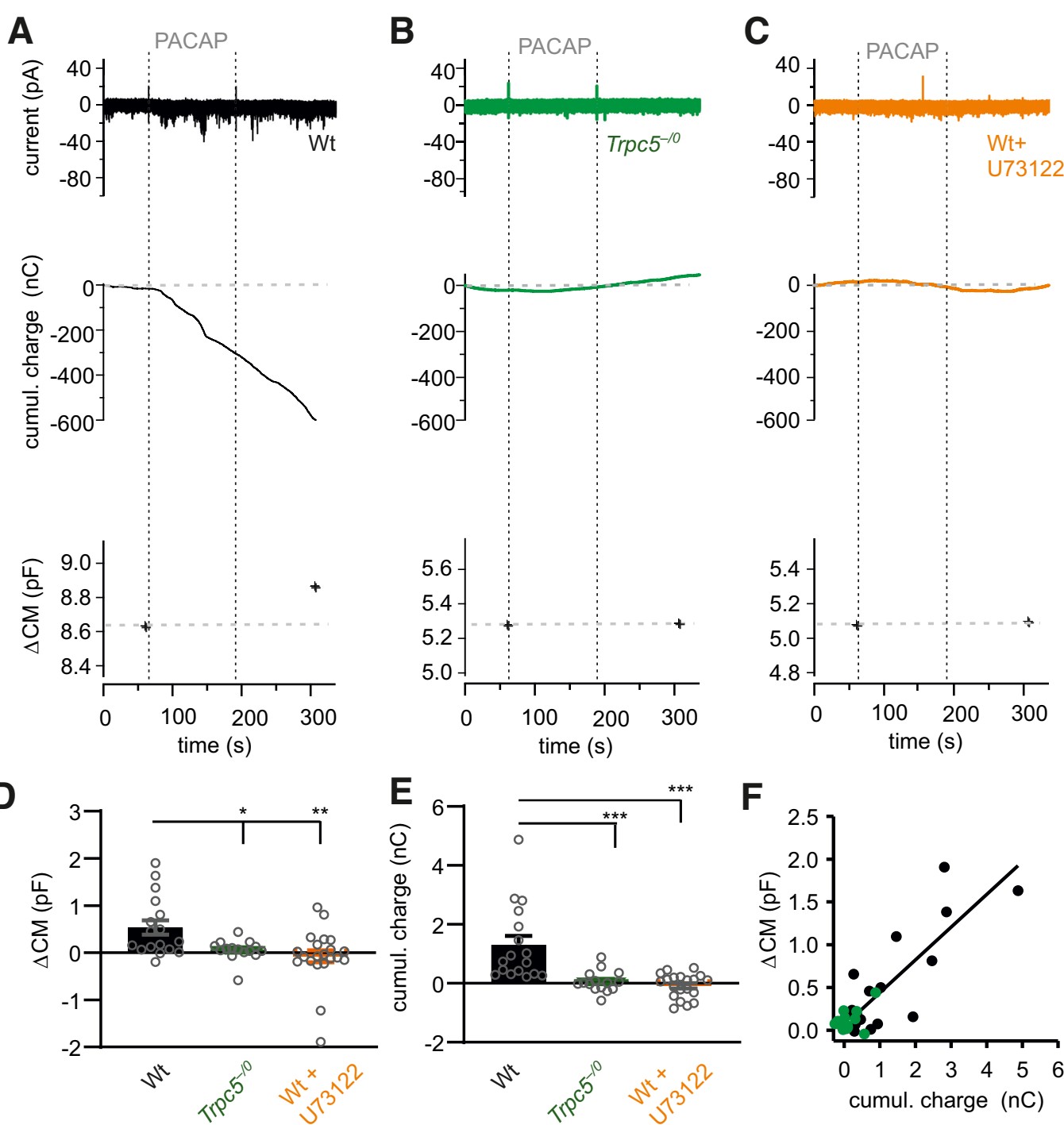

Figure 7. **PACAP evokes TRPC5 channel mediated inward currents in a PLC-dependent manner.**

(**A**) Exemplary recordings (perforated-patch configuration) of a PACAP-evoked inward current in a wt cell (upper panel), the corresponding cumulative charge plot (middle panel) and membrane capacitance measurement (lower panel). The capacitance increase (ΔCM) in response to the stimulation reflects dense core vesicle exocytosis. (**B**) PACAP fails to evoke any inward current or secretion in *Trpc5⁻/⁰* cells. (**C**) The PLC–inhibitor U-73122 (10 μM) blocks the PACAP induced current and any secretion in wt cells. (**D**) Genetic loss of TRPC5 or inhibition of the PLC prevents any change in CM (ΔCM) seen in wt cells upon PACAP application. Data were collected from wt, $n = 17$, *Trpc5⁻/⁰*, $n = 15$ $p = 0.0377$ and wt+U-73122, $n = 21$ $p = 0.0026$ from 3 independent preparations. (**E**) The cumulative charge transfer is abolished in the absence of TRPC5 or after PLC inhibition, data were collected from wt, $n = 17$, *Trpc5⁻/⁰*, $n = 15$, $p = 0.00002$ and wt+U-73122, $n = 20$, $p = 0.000011$ from 3 independent preparations. (**F**) Changes in membrane capacitance (indicative of exocytosis) correlate with the charge influx evoked by PACAP-38, data were collected from wt, $n = 17$, *Trpc5⁻/⁰*, $n = 15$. Bar graphs are displayed as mean ± s.e.m., Mann–Whitney U test, *$p < 0.05$, **$p < 0.01$, ***$p < 0.001$. Source data are available online for this figure.

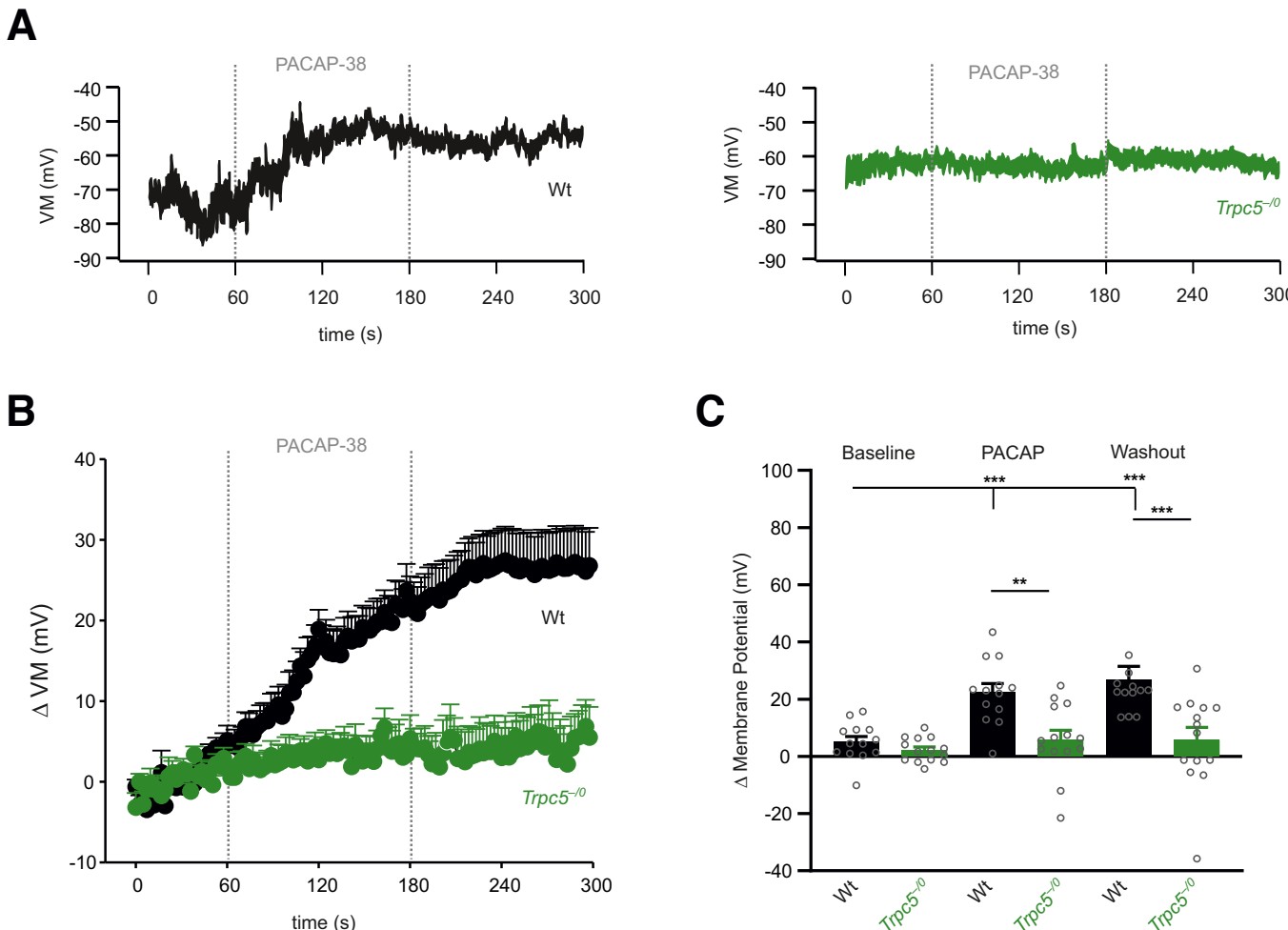

**Figure 8.  PACAP induces long lasting TRPC5 mediated depolarization of the membrane potential.**

(A) Exemplary recordings (perforated-patch and current-clamp configuration) of a PACAP-evoked membrane depolarization in wt cells (left panel). No changes in VM could be detected in $Trpc5^{-/0}$ cells (right panel). (B) Time course of the average PACAP-evoked depolarization of the membrane potential, mean ± s.e.m. (C) PACAP leads to a significant increase in the VM in wt cells but not in $Trpc5^{-/0}$ cells. Data were collected from wt, $n = 13$ and $Trpc5^{-/0}$, $n = 14$ cells from 3 independent preparations. One-way ANOVA Kruskal Wallis for statistical testing within the group (wt, baseline vs. PACAP $p = 0.001$; wt baseline vs. washout $p = 0.0001$) and Mann–Whitney U-test between wt and $Trpc5^{-/0}$ (wt vs. $Trpc5^{-/0}$ PACAP, $p = 0.00275$; Washout, $p = 0.000528$; **$p < 0.01$, ***$p < 0.001$, bar graphs are presented as mean ±s.e.m. Source data are available online for this figure.

$Trpc5^{fx/0}$;DBH-Cre⁻ and $Trpc5^{fx/0}$;DBH-Cre⁺ mice. Also under these conditions, we observed no differences in either glucocorticoid or glucagon plasma concentrations (Appendix Fig. S3A–D). On the contrary, the rise in plasma adrenaline levels that was evoked by hypoglycemia was blunted by TRPC1/4/5/6 deletion. This demonstrates that TRPC channels are key players in the counter regulation via the HSA axis. After pinpointing this defect to the TRPC5 subtype, we obtained additional evidence regarding the importance of TRPCs in HSA-mediated actions. During ITTs, systemic adrenaline supplementation prevented the more pronounced decline in blood glucose levels in the TRPC5-deficient mice. Interestingly, Gao et al reported no major alterations in ITTs (Gao et al, 2017) in a TRPC5-deficient mouse with a background and deletion (129/SvImJ, exon 5) that differs from our model (C57B6/N, exon4) (Riccio et al, 2009). The genetic background and gene ablation method as well as the used insulin dose and fasting duration are potential explanations for such

discrepancies. To verify the hypoglycemia phenotype in an independent TRPC5-deficient mouse model, we analyzed the $Trpc5^{-/0}$_LB line (mixed 129EvSv/C57BL/6J background) which displayed significantly lower blood glucose levels and a lack of adrenaline elevation. The pivotal role of TRPC5 in hypoglycemia-evoked autonomic counter-regulation and adrenaline homeostasis was further supported by experiments with the TRPC5-specific agonist Englerin A (EA) (Beck et al, 2017). Notably, systemic EA application evoked a severe plasma adrenaline rise within 5 min that was absent in the $Trpc5^{-/0}$ animals.

The counter regulation via the HSA axis triggered by hypoglycemia is initiated by the activation of glucose-excited (GE) and glucose-inhibited (GI) neurons (Steinbusch et al, 2015) in the hypothalamus and in the RVLM (Ritter et al, 2011), both of which project to the splanchnic nerve that innervates the chromaffin cells of the adrenal medulla. Insulin receptor stimulation activates TRPC5 channels in POMC neurons of the central

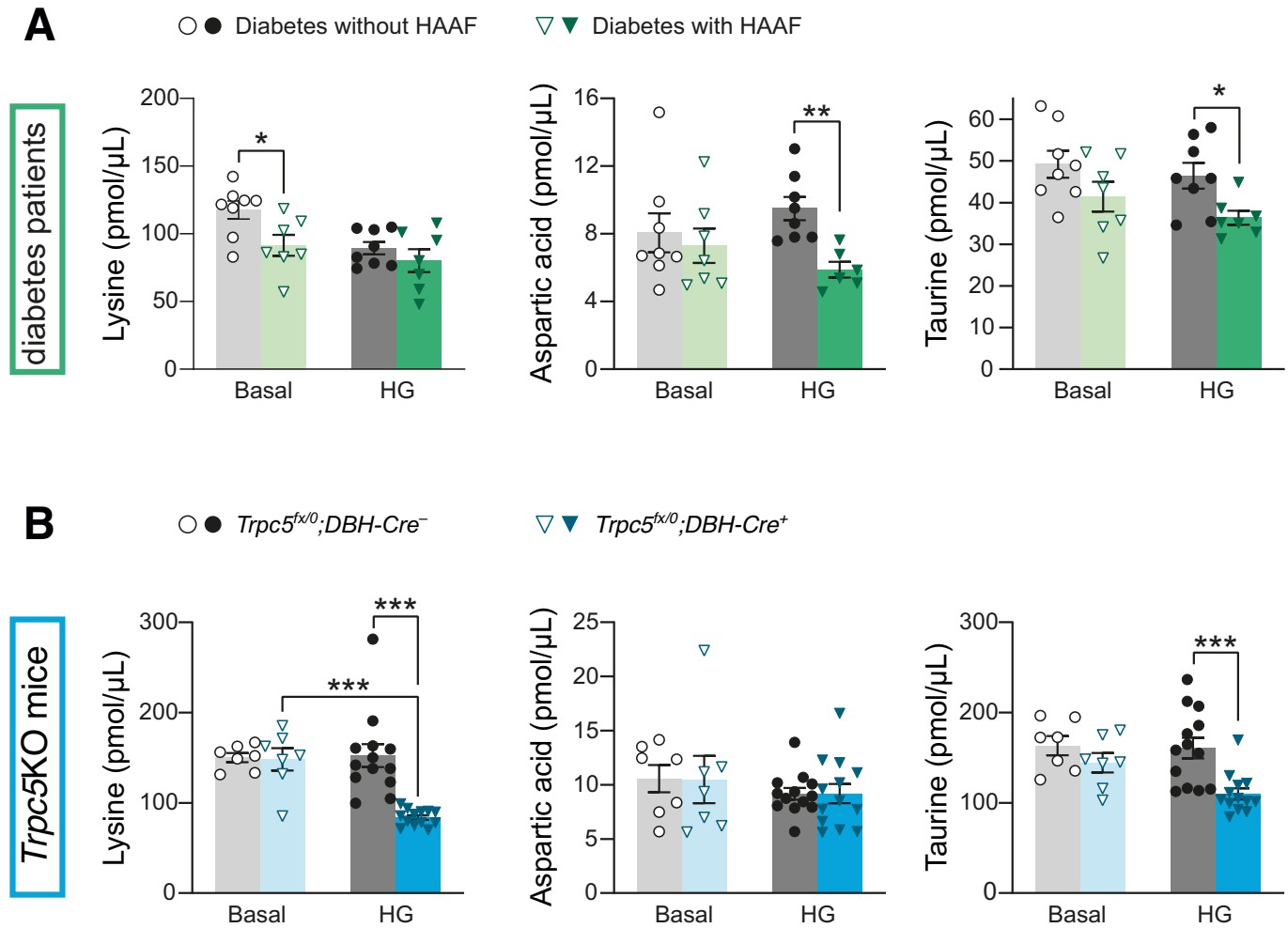

**Figure 9.  Metabolic parallels between TRPC5 deficiency and HAAF.**

(A) Plasma levels of lysine, aspartic acid, and taurine (from left to right) for diabetic patients diagnosed with HAAF before (basal) and during controlled hypoglycemia (HG; 60–70 mg/dL). $n = 7$ for HAAF patients, $n = 8$ for diabetes patients without HAAF. Lysine, basal, $p = 0.0230$; Aspartic acid, HG, $p = 0.00169$; Taurine, HG, $p = 0.0166$. (B) Plasma levels of lysine, aspartic acid, and taurine 30 min after the injection of saline (basal) or insulin (HG; 2.25 U/kg insulin, i.p.) for mice with *Trpc5* deletion in catecholaminergic cells. *Trpc5$^{fx/0}$;DBH-Cre$^+$*: $n = 13$ for insulin, $n = 7$ for saline; *Trpc5$^{fx/0}$;DBH-Cre$^-$*: $n = 13$ for insulin, $n = 7$ for saline). Lysine, HG, $p = 2.08 \times 10^{-5}$; Lysine, *Trpc5$^{fx/0}$;DBH-Cre$^+$*, $p = 2.24 \times 10^{-6}$; Taurine, HG, $p = 0.000715$. HG, hypoglycemia. (A, B) Mean ± s.e.m., *$p < 0.05$, **$p < 0.01$, ***$p < 0.001$, two sample *t* test. Source data are available online for this figure.

nervous system (Qiu et al, 2014), giving rise to the hypothesis that TRPC5 channels in gluco-regulatory CNS neurons might be similarly directly activated by insulin. However, the intranasal administration of insulin, which allows a more direct access to the cerebrospinal fluid than systemic or intraperitoneal application, did not provoke hypoglycemia (Gray et al, 2014). Congruently, intracerebroventricular injection neither affected the blood glucose levels nor the activity of the sympathetic nerves in the adrenal gland (Muntzel et al, 1994). To trigger the autonomous nervous system independently of hypoglycemia, we challenged the mice systemically with LPS (i.p.) to induce stress through systemic inflammation (Seemann et al, 2017), or with histamine (i.v.) which leads to a substantial drop in body temperature (Bugajski and Zacny, 1981) and subsequently to an increase in plasma catecholamines (Shimizu et al, 2006). We show that the plasma adrenaline levels after LPS (Appendix Fig. S2A) and histamine

injections (Appendix Fig. S2B), respectively, are largely attenuated in *Trpc5*-deficient mice compared to controls.

To pinpoint the cell type in which TRPC5 operates in autonomous counter regulation, we generated mice in which the *Trpc5* gene was inactivated in cholinergic (*Trpc5$^{fx/0}$;ChAT-Cre$^+$*) or catecholaminergic cells (*Trpc5$^{fx/0}$;DBH-Cre$^+$*). While hypoglycemia counter-regulation was unaffected in the *Trpc5$^{fx/0}$;ChAT-Cre$^+$* animals, the male and female *Trpc5$^{fx/0}$;DBH-Cre$^+$* mice exhibited more severe hypoglycemia paralleled by significantly lower plasma adrenaline levels. The results rule out a contribution of TRPC5 in the cholinergic neurons of the splanchnic nerve and render a role of TRPC5 in glucose-sensing GE/GI neurons in the hypothalamus implausible since these neuronal circuits are not considered catecholaminergic. Conversely, the results assign an essential function to TRPC5 for counter regulation in catecholaminergic cells. Hypoglycemia-activated cells of the autonomous nervous

system include neurons in the NTS (Adachi et al, 1995; Balfour et al, 2006) and LC (Morilak et al, 1987). Counter regulation leading to the restoration of euglycemia essentially involves the RVLM neurons of the hindbrain and the chromaffin cells of the adrenal medulla (Ritter et al, 2011; Verberne et al, 2014; Zhao et al, 2017). The autonomic failure in the *Trpc5fx/0;DBH-Cre+* mice, as well as the poor co-localization of TRPC5 and TH (as a marker of catecholaminergic cells) in the NTS, make an impact of TRPC5 in NTS neurons in this process unlikely. The AAV2-mediated expression of Cre recombinase in catecholaminergic RVLM neurons did not evoke the phenotype observed in *Trpc5* KO animals during hypoglycemia, making a contribution of TRPC5 in this brain region unlikely. As only 59% of the TH-positive RVLM neurons expressed Cre, a functional involvement of TRPC5 in central counter-regulatory cascades cannot entirely be excluded. Nevertheless, our profiling of TRPC5 activity throughout the autonomic regulatory pathway renders important signaling hubs in the CNS unlikely.

## TRPC5 channels govern sustained adrenaline secretion from chromaffin cells

Importantly, the systematic examination of the triple deficient *Trpc1/4/5−/−*, the single *Trpc5* KO, and the cell type-specific *Trpc5* KO mice (*Trpc5fx/0;DBH-Cre+*) revealed comparable physiological deficits and thus allowed us to pinpoint not only the relevant TRPC subtype but also chromaffin cells as the catecholaminergic cells of the peripheral autonomous nervous system as the determinant site of action. In pursuing this strategy, we identified TRPC5 channels are potent regulators of the cytosolic $Ca^{2+}$ concentration in chromaffin cells and can be activated by either ACh or the neuropeptide PACAP, both of which are physiologically released at the splanchnic-adrenal synapse. Indeed, PACAP application evokes TRPC5-mediated currents in chromaffin cells from Wt but not *Trpc5−/0* mice. Consequently, stimulation of chromaffin cells with PACAP or muscarine strongly enhanced granule exocytosis in a TRPC5-dependent fashion, as judged from our amperometric recordings. Both, genetic loss of TRPC5 in chromaffin cells or its direct pharmacological inhibition diminished the secretion response, indicating that acute loss of TRPC5 activity rather than potential developmental adaptions in the absence of TRPC5 channels are responsible for the observed phenotype. Likewise, the TRPC5 blocker C31 efficiently blocked the hypoglycemia-evoked adrenaline elevation in vivo. Moreover, the absence of an EA-induced increase in plasma adrenaline of *Trpc5−/0* mice suggests an effect on TRPC5-expressing cells outside the CNS-most likely in chromaffin cells of the adrenal medulla, as EA was detected only in blood plasma, but not in brain after intraperitoneal application (Cheung et al, 2018). In summary, our organismal analyses together with our ex vivo studies provide striking parallels and converging lines of evidence that TRPC5 activity at the level of chromaffin cells constitutes a crucial step in the homeostatic counter-regulation of insulin-induced hypoglycemia. Our results are consistent with previous observations (Eiden et al, 2018; Harada et al, 2015; Kuri et al, 2009) and also provide an attractive mechanistic explanation of how muscarine and PAC1 receptors at the adrenomedullary synapse can trigger sustained secretion of adrenaline.

Muscarinic and PAC-1 receptors are activated by acetylcholine and PACAP, which are released from small synaptic and large-dense core vesicles by the splanchnic nerve, respectively. With this combination of diverse receptors, chromaffin cells are well suited to translate the different sympathetic activity patterns into a sustained secretion response from chromaffin cells meeting the requirements to maintain euglycemia under stress conditions. Congruent with this result, the hypoglycemia-evoked plasma adrenaline rise is significantly reduced in PACAP-deficient mice (Hamelink et al, 2002). We also show that pharmacological inhibition of the phospholipase C (PLC) abolishes secretion from chromaffin cells evoked by PACAP stimulation (Appendix Fig. S6). Yet, being aware of the observation that exocytosis itself is potentiated by PLC activation in a Munc13-1-dependent manner (Bauer et al, 2007), we directly recorded TRPC5 channel activity (Fig. 7), which was tightly coupled with a robust increase in membrane capacitance, indicative of secretion. Both cellular responses were found to be absent in the *Trpc5−/0* mice and to be sensitive to PLC inhibition. Thus, PLC activation mediates an essential, intermediate step in the PAC1 receptor-TRPC5 channel coupling, providing new mechanistic insight into the signaling pathway leading to adrenaline secretion and euglycemia.

Our findings agree with, and extend, previous observations (Kuri et al, 2009) by showing that PACAP induces a long-lasting depolarization of the membrane potential which remains below threshold and depends on TRPC5 channel activity. Given an average plasma membrane resistance of 2.5 GOhm for a chromaffin cell, the magnitude of membrane depolarization (~20 mV, Fig. 8) is well explained by the mean TRPC5 current amplitude of about 8 pA (cumulative charge/time = 1.4 nC/180 s) recorded in the voltage-clamp experiments (Fig. 6). This suggests that the TRPC5-mediated inward current is the predominant source for the observed voltage change. Nevertheless, coactivation of other voltage-dependent conductances in response to TRPC5-mediated depolarization, particularly of low-voltage activated T-type $Ca^{2+}$ channels (Hill et al, 2011; Mahapatra et al, 2012), cannot be completely ruled out as an additional pathway for calcium influx into chromaffin cells. In any case, our experiments show that TRPC5 activity suffices to stimulate the secretion of chromaffin cells. Notably, onset and time course of the PACAP-evoked $Ca^{2+}$ rise and the TRPC5 current activation, acting at the second time scale, compares well with the kinetics of the Gq/PLC pathway (including $PIP_2$ hydrolysis) (Jensen et al, 2009) rather than with timing of the cAMP/PKA dependent upregulation of L-type channels (Cav1.2, Cav1.3), which requires minutes to reach maximal rates (Mahapatra et al, 2012). For synaptic transmission, a comparable role of TRPC5 was recently uncovered at presynaptic sites in hippocampal neurons (Bröker-Lai et al, 2017; Schwarz et al, 2019). Here, TRPC5 channels establish an independent $Ca^{2+}$ entry pathway that functionally couples to the $Ca^{2+}$ influx mediated by voltage-gated $Ca^{2+}$ channels and prolongs the presynaptic $Ca^{2+}$ rise. Overall, our experiments describe for key adrenomedullary neurotransmitters, a previously unknown signaling pathway that activates TRPC5 in a receptor-operated PLC-dependent manner to promote sustained release of adrenaline from chromaffin cells.

In this context it stands to reason that TRPC1 channels have also been associated with the regulation of catecholamine release (Marom et al, 2011), which seems to contradict our observations on the specific involvement of TRPC5 channels. While TRPC1, TRPC4, and TRPC5 form heteromultimeric channels (Bröker-Lai

et al, 2017) only homomeric TRPC5, but not TRPC1 channels have a strong impact on synaptic plasticity (Schwarz et al, 2019), a notion that agrees with observations in heterologous expression systems in which no evidence for functional homomeric TRPC1 channels was found (Storch et al, 2012; Strubing et al, 2001). A possible solution for the apparent contradiction is provided by our recent observation showing that knock out of TRPC1 in brain also decreased the amounts of TRPC4 and TRPC5 by 56% and 67%, respectively (Kollewe et al, 2022). Thus, genetic loss of TRPC1 in chromaffin cells may similarly reduce TRPC5 expression.

## Sympatho-adrenal failure as a commonality between TRPC5-deficient mice and HAAF syndrome in diabetic patients

HAAF (hypoglycemia-associated autonomic failure) is a clinical syndrome of defective glucose counter regulation to episodes of hypoglycemia in people with established type 1 or advanced type 2 diabetes. It is characterized by a combination of compromised physiological defenses against hypoglycemia, i.e., a decrease in glucose lowering insulin, and an increase in the glucose raising hormones glucagon and adrenaline. The attenuated defense causes hypoglycemia unawareness, which drastically increases the risk of recurrent severe hypoglycemia (Cryer, 2013). The mechanisms how and why adrenaline secretion is impaired in HAAF patients is not well understood on the molecular level. Our results show a striking parallel in sympatho-adrenal failure after the occurrence of hypoglycemia between *Trpc5* KO mice and in HAAF patients. One might speculate that pertubations in the TRPC5 activity are involved in the pathophysiology of HAAF. However, the defect in autonomic counter regulation in *Trpc5* KO mice is not specifically triggered by induction of hypoglycemia but is also observed in other forms of stress and in HAAF patients, a reduced sympathetic activity is observed (Cryer, 2013), which is upstream of the chromaffin cells. Nevertheless, follow-up studies are needed to show more directly a contribution of the TRPC5-dependent mechanism to sympatho-adrenal failure. In this case, TRPC5 channel agonists such as englerin A (EA) or other more tolerable agonists, such as tonantzitlolone, riluzole, or BTD (Minard et al, 2019) bear potential as symptomatic therapy, acting downstream of the impaired sympathetic activity, for the still-incurable long-term diabetic complication.

A particular hazardous aspect of HAAF is that the glucose concentration in the blood may drop to life-threatening levels (Kalra et al, 2013), and treatment of HAAF patients relies only on prevention of hypoglycemia episodes. Until now there is also a lack of specific approaches for their prediction. We looked for similarities in the metabolite signature in an unbiased metabolome analysis in plasma. Among the metabolites analyzed, we specifically observed the decrease in taurine levels as a coincidence between *Trpc5* KO mice and HAAF patients under hypoglycemia. Taurine is one of the most abundant free amino acids in mammalian tissues (Ripps and Shen, 2012). It can be synthesized endogenously from cysteine or methionine, for which we did not observe any differences in the HAAF patients, arguing against a defect in taurine synthesis. Several preclinical studies emphasize cytoprotective effects of taurine and therapeutic ameliorations in experimental and human diabetes including neurotransmission (Sirdah, 2015), e.g., by attenuating hyperalgesia and abnormal $Ca^{2+}$ signaling in

sensory neurons (Li et al, 2005). The mechanisms that lead to changes in taurine plasma levels are very complex and may differ between *Trpc5* KO mice and in HAAF patients as taurine metabolism is influenced by many metabolic pathways (Hayes, 1988; Ripps and Shen, 2012). In addition, a reduction in plasma taurine levels can be observed when it is used as a source of energy production during severe hypoglycemic conditions (Battezzati et al, 2000). Plasma taurine levels, can be determined by routine procedures, however, it needs to be confirmed in further studies with larger patient cohorts whether the decrease in taurine plasma levels could indeed serve as a diagnostic marker with sufficient sensitivity and specificity in HAAF patients or for certain HAAF subgroups.

## Conclusion

Our study identifies for the first time TRPC5 channels in adrenal chromaffin cells as critical regulators of calcium-dependent adrenaline secretion. Reduced hypoglycemia-evoked adrenalin secretion, as observed in *Trpc5* KO deficient mice, contributes significantly to hypoglycemia-associated autonomic counter regulation. The similarities in sympatho-adrenal insufficiency between *Trpc5* KO mice and HAAF patients identified for the first time, represent a new avenue for the development of new TRPC5-associated diagnostic and treatment options for the long-term diabetic complication HAAF.

## Methods

### Study subjects

#### Animals

All experiments were performed on 2 to 3 months old age-matched male or female mice, to limit variability between animals. Mice were housed in a 12-h light-dark cycle, with a relative humidity between 56 and 60%, a 15-times air change per hour and room temperature of $22\,°C +/-2\,°C$. They were kept in conventional cages type II or type II long provided with animal bedding LTE E-001 (ABBEDD, Germany) and tissue papers as enrichment. Standard autoclaved food (Rod 16 or 18, Altromin, Germany) and autoclaved water were available to consume ad libitum. The mice were randomly assigned to an experimental group and, where possible, the experimental scientist was unaware of the group allocations. The mouse lines $Trpc1/4/5/6^{-/-}$ and $Trpc1/4/5^{-/-}$ were generated by intercrossing mice of the lines $Trpc1^{-/-}$ (MGI:3764882) (Dietrich et al, 2007), $Trpc4^{-/-}$ (MGI:2387666) (Freichel et al, 2001), $Trpc5^{-/0}$ (MGI:5474572) (Xue et al, 2011), and $Trpc6^{-/-}$ (MGI:3623135) (Dietrich et al, 2005), respectively. Before generating lines with deletions in multiple genes, all *Trpc* single knockout mouse lines had been backcrossed to the defined C57BL/6N strain for at least five generations. $Trpc1/4/5/6^{-/-}$ and $Trpc1/4/5^{-/-}$ mice were compared to C57BL/6N mice, which were obtained from Charles River and housed under the same conditions as the knockout (KO) animals. $Trpc5^{-/0}$ animals were also compared to litter-matched controls, as indicated. For the validation of our results, we used an independently generated *Trpc5* KO, $Trpc5^{-/0}\_LB$ mice (MGI:5474583) with a 129EvSv/C57BL/6J background

(Phelan et al, 2013). Here, we compared litter-matched animals. Mouse lines with a cell type-specific deletion of *Trpc5* were generated by crossing *Trpc5^{fx/0}* mice carrying a floxed exon 4 (Xue et al, 2011), with the following mouse lines: *ChAT* (choline acetyltransferase)-*Cre* (MGI:5475195; JAX stock #006410) (Lowell et al, 2006; Rossi et al, 2011) for expression in cholinergic neurons and *DBH* (dopamine-β-hydroxylase)-*Cre* (MGI:4355551) (Parlato et al, 2007) for expression in catecholaminergic cells. For these conditional *Trpc5* KO animals, we compared *Trpc5^{fx/0};Cre-negative* (*Cre^−*) and *Trpc5^{fx/0};Cre-positive* (*Cre^+*) male or *Trpc5^{fx/fx};Cre-negative* (*Cre^−*) and *Trpc5^{fx/fx};Cre-positive* (*Cre^+*) female mice. For visualizing *Trpc5* expression in TH-positive cells, a mouse line, in which the Cre recombinase was expressed under the control of the *Trpc5* promotor (*Trpc5-IRES-Cre*) (Wyatt et al, 2017), was crossed to an enhanced ROSA26-floxed-stop-tau-GFP mouse (*eR26-τGFP*) to identify *Trpc5*-expressing cells by visualization of GFP (Schwarz et al, 2019). For detection of TRPC5 proteins in immunostainings the *Trpc5^{−/0}* (MGI:5474572) strain and their *Trpc5^{+/0}* littermates were used. Male mice were mainly used to identify and understand the intricacies of the studied pathway with minimal confounding sources of variability introduced by hormonal fluctuations and reproductive cycles. We confirmed our key findings in female mice in separate experiments, and could not observe any sex-dependent differences.

All experimental procedures, including the blinding and randomization procedures, were performed in accordance with the ARRIVE guidelines, the welfare regulations and ethical guidelines as approved by the local governing body (Regierungspräsidium Karlsruhe, Germany, approval numbers 35-9185.81/G-219/19; Amtstierärztlicher Dienst, Saarland Germany, approval number: 2.4.1.1-CIPMM and GB3-2.4.7.1). Regierung von Oberbayern, Veterinärwesen) TVA: 55.2-2532.Vet_02-21-133.

### Patients with type 1 diabetes

For metabolome analysis, plasma of patients with type 1 diabetes with and without HAAF was provided by the University Hospital Heidelberg (Department of Medicine I: Endocrinology and Clinical Chemistry), sex or gender identity of the patients was not an inclusion or exclusion criterion (Appendix Fig. S10). Blood samples were taken from the patients before and at 4 time points during a hyperinsulinemic clamp study with an additional hypoglycemic phase (blood glucose levels: 60–70 mg/dL) (Groener et al, 2019). The metabolite measurements reported in this manuscript occurred on plasma samples, after all samples were collected and stored in a biobank and falls under the plasma collection and biobanking included in the study protocol for this clinical study, which primary outcome (asprosin dynamics during hypoglycemia) is described elsewhere (Groener et al, 2019). The study was initially planned as a pilot study with 10 patients (5 with and 5 without HAAF) and has been positively evaluated by the local ethics committee in accordance with the Declaration of Helsinki (Ethikkommission der Medizinischen Fakultät der Universität Heidelberg, ethics number S-550/2016) and registered on ClinicalTrials.gov with Identifier NCT03358121. An amendment to include more patients was approved by the local ethics committee (Ethikkommission der Medizinischen Fakultät der Universität Heidelberg, ethics number S-381/2019). The study protocols conform to the principles set out in the Belmont Report from the U.S. Department of Health and Human Services. Every participant has given their written informed consent. Eight female and 7 male subjects participated to the study, sex was self-reported. Data was collected from $n = 7$ for HAAF patients, $n = 8$ for diabetes patients without HAAF.

## Murine diabetes model

For the induction of diabetes, animals received intraperitoneal injections of 60 mg/kg body weight streptozotocin (STZ; Sigma-Aldrich) on five consecutive days. Resultant diabetic hyperglycemia was maintained in the range of 300–500 mg/dL (ACCU-CHEK® Aviva, Modell: NC, mg/dL; Roche) with subcutaneous injections of the long-acting insulin glargine (Lantus®, Sanofi-Aventis) twice weekly, according to the following regimen: blood glucose (BG) $\geq 400$, $<450$ mg/dL $\rightarrow 1$ U insulin; BG $\geq 450$, $<500$ mg/dl $\rightarrow 2$ U; BG $\geq 500$, $<600$ mg/dL $\rightarrow 3$ U; BG $\geq 600$ mg/dL $\rightarrow 4$ U. Two weeks after the start of the controlled insulin treatment, the protocol was adapted for the *Trpc*-knockout group. Now, they received ¼ U insulin only when their blood glucose levels were 600 mg/dL or above.

## Metabolic tests

During the metabolic tests, mice were food-deprived with water ad libitum. After 1 h of fasting (exception: glucose tolerance test (GTT): overnight), the animals' reference blood glucose levels (0 min) were measured. All glucose measurements were conducted with a Roche ACCU-CHEK® Aviva glucometer and by taking 2–3 μL of blood through incision of the tail tip. The applied substances were diluted in isotonic saline solution (0.9% NaCl), unless otherwise stated. Groups of animals, which were injected with plain solvent according to their body weights, served as negative controls.

For glucose tolerance tests (GTTs), mice were fasted overnight (for 16 h) before they received an intraperitoneal (i.p.) injection of 2 g/kg body weight glucose (B. Braun). The blood glucose concentrations were determined 0, 15, 30, 60, and 120 min after the glucose administration.

For insulin tolerance tests (ITTs), the reference blood glucose measurement was followed by an i.p. injection of 2.25 U/kg body weight of human insulin (Insuman Rapid®, Sanofi-Aventis). Blood glucose levels were determined 0, 15, 30, 45, and 60 min—or for a long-term experiment—0, 60, 90, 120, and 150 min after the insulin injection.

To study the time course of the blood glucose levels upon glucagon administration, mice were injected intraperitoneally with 16 μg/kg body weight of glucagon (GlucaGen®, Novo Nordisk). Blood was taken 0, 15, 30, 45, 60, and 90 min after the glucagon application.

To study the time course of the blood glucose levels upon cortisol administration, animals received an intraperitoneal injection of 100 mg/kg body weight of cortisol (Hydrocortison, Rotexmedica). Blood glucose concentrations were measured 0, 15, 30, 45, 60, 90, and 120 min after the cortisol application.

For the adrenaline rescue experiment, the ITT protocol was modified as follows: The insulin administration was followed (2–3 min later) by an intravenous injection of adrenaline (0.6 mg/kg body weight) under a short-term isoflurane anesthesia. Blood glucose levels were measured 10, 20, 30, and 45 min after the injection of adrenaline. The reference groups received saline, instead of adrenaline, intravenously after the insulin application.

To study the effects of LPS on adrenaline plasma levels, the animals received 10 mg/kg Lipopolysaccharides from E.coli 055:B5 (Sigma L2880) through i.p. injection at a concentration of 3.3 mg/mL in saline. Plasma samples for adrenaline concentration determination were taken 360 min after LPS injection.

For the histamine challenge test, a temperature transponder (IPTT-300, BMDS) was implanted subcutaneously on the dorsal midline during isoflurane anesthesia 24 to 96 h prior to the experiment. Prior to the histamine challenge test, the mice were fasted for 1 h and the effects of histamine injection on blood glucose and adrenaline plasma levels were tested after i.v. injection of 30.66 mg/kg histamine. Effective histamine administration was confirmed with a decrease of 2 °C in body temperature (Bugajski and Zacny, 1981). Blood glucose was measured 0, 15, and 30 min after histamine injection and at 30 min a plasma sample was taken for adrenaline concentration determination.

To test the effect of the TRPC4/TRPC5 agonist Englerin A (Roth) on the blood glucose levels, 5 mg/kg body weight of Englerin A were injected intraperitoneally. Englerin A was dissolved in 5% Ethanol, 10% polyethylene glycol 300, 5% Cremophor® EL, and 80% PBS (Carson et al, 2015). Glucose concentrations in the blood were determined 5 min after Englerin A administration.

The effect of the TRPC4/TRPC5 antagonist C31 on the blood glucose levels was tested during an ITT. To this end, $2 \times 30$ mg/kg body weight of C31 were applied orally by gavage the night before and on the morning of the ITT 2 to 4 h before the insulin injection. C31 was suspended in 0.5% methyl cellulose. Blood glucose levels were measured 0, 15, 30, 45, and 60 min after the insulin application. The dose of C31 and the duration of the experiment were planned according to pharmacokinetic analysis. The pharmacokinetic profile of C31 administered at 30 mg/kg p.o. bid in plasma was as follows: $t_{1/2} = 2.55$ h; $t_{max} = 1$ h; $c_{max} = 1.9$ μM; AUC = 5232 h × ng/mL; F% oral bioavailability = 38%.

The plasma levels of glucagon and corticosterone were determined during the steady state phase at 50 mg/dL of a hyperinsulinemic hypoglycemic clamp experiment. The mice (3.5-month-old $Trpc5^{fx/0};DBH\text{-}Cre^+$ and $Trpc5^{fx/0};DBH\text{-}Cre^-$ male littermates) received a permanent jugular vein and carotid artery catheters (Instech Laboratories) under MMF (Midazolam (5 mg/kg), Medetomidin (0.5 mg/kg), Fentanyl (0.05 mg/kg) i.p.) anesthesia. The left common carotid artery was catheterized for sampling blood and the right jugular vein was catheterized for infusion (Ayala et al, 2011). The free catheter ends were tunneled under the skin to the back of the neck and connected with a two channel vascular access button™ (Instech Laboratories). After a post-surgical recovery period of 6 or 7 days, hyperinsulinemic hypoglycemic clamp was performed in conscious mice. Food access was restricted one hour before clamp. Plasma levels of glucagon and corticosterone were determined at t0 before insulin (20 mU/kg/min) and variable glucose infusion starts and during the steady state phase (t1) of the hyperinsulinemic hypoglycemic clamp at ~50 mg/dL. Blood glucose levels were monitored periodically every 10 min till steady state was reached.

## Hormone and metabolite measurements

To measure the hormone levels in plasma, at least 500 μL blood was taken per mouse, puncturing the buccal venous plexus directly after the last blood glucose measurement. Subsequently, mice were killed by cervical dislocation. For the detection of catecholamines and corticosterone, blood was collected in 1.5-mL tubes, containing 50 μL of EGTA-GSH. For glucagon and metabolites, 1.3 mL Sarstedt micro tubes were used, whose walls were covered with EDTA. Within 30 min, the blood was centrifuged for 10 min at $6150 \times g$. Plasma was transferred to fresh test tubes and stored at −20 °C until assayed.

The concentrations of the catecholamines adrenaline, noradrenaline, and dopamine were determined by HPLC with amperometric detection (reagent: Chromsystems, Germany). For corticosterone measurements, a competitive radioimmunoassay was employed, using tritiated corticosterone. The analyses of catecholamines and corticosterone were carried out by the central laboratory of Heidelberg University Hospital, using the same procedures validated for routine diagnostic analysis.

Glucagon levels were measured by double antibody radioimmunoassay (IBL international) at Labor Limbach (Heidelberg) or with the mouse glucagon ELISA kit (Crystal Chem) for plasma samples after the hyperinsulinemic hypoglycemic clamp experiments.

The plasma amino acid, TCA cycle metabolites and fatty acids in plasma samples from HAAF patients and $Trpc5^{fx/0};DBH\text{-}Cre^+$ mice in control conditions and during evoked hypoglycemia (Figs. 9, Appendix Fig. 8, 9) were quantified by the Metabolomics Core Technology Platform of the Centre for Organismal Studies (COS), University of Heidelberg.

Free amino acids were analyzed after 1:30 dilution of plasma samples with ice-cold 0.1 M HCl to inactivate and precipitate proteins. Samples were incubated for 15 min on ice. The resulting extracts were centrifuged for 10 min at 4 °C and $16,400 \times g$ to remove proteins and cell debris. Amino acids were derivatized with AccQ-Tag reagent (Waters) and determined as previously described (Weger et al, 2016). Determination of organic acids was adapted from Uran et al (Uran et al, 2007), briefly, plasma was diluted 1:10 with ultra-pure water. 0.45 mL ice-cold methanol was added to 0.1 mL diluted plasma and incubated on ice for 15 min. After centrifugation (10 min; 4 °C; $20,000 \times g$) to remove precipitated proteins, 50 μL of the supernatant was mixed with 25 μL 140 mM 3-Nitrophenylhydrazine hydrochloride (Sigma-Aldrich), 25 μL methanol and 100 μL 50 mM Ethyl-3-(3-dimethylaminopropyl) carbodiimide hydrochloride (Sigma-Aldrich) and incubated for 20 min at 60 °C. Separation was carried out on the above described UPLC system coupled to a QDa mass detector (Waters) using an Acquity HSS T3 column (100 mm × 2.1 mm, 1.8 μm, Waters) which was heated to 40 °C. Separation of derivates was achieved by increasing the concentration of 0.1% formic acid in acetonitrile (B) in 0.1% formic acid in water (A) at 0.55 mL × min⁻¹ as follows: 2 min 15% B, 2.01 min 31% B, 5 min 54% B, 5.01 min 90% B, hold for 2 min, and return to 15% B in 2 min. Mass signals for the following compounds were detected in single ion record (SIR) mode using negative detector polarity and 0.8 kV capillary voltage: Lactate (224.3 $m/z$; 25 V CV (cone voltage)), malate (403.3 $m/z$; 25 V CV), succinate (387.3 $m/z$; 25 CV), fumarate (385.3 $m/z$; 30 V), citrate (443.3 $m/z$; 10 V), pyruvate (357.3 $m/z$; 15 V) and ketoglutarate (550.2 $m/z$; 25 CV). Data acquisition and processing was performed with the Empower3 software suite (Waters).

For the free fatty acid determination, frozen plasma samples (50 μL) were extracted with 180 μL 100% methanol (HPLC-grade)

for 15 min at 70 °C with vigorous shaking. After the addition of 100 μL 100% chloroform (HPLC-grade) containing 20 mg/mL C17:0 (Heptadecanoic acid; Sigma H3500) as internal standard, samples were shaken for 5 min at 37 °C. To separate polar and organic phases, 200 μL HPLC-grade water were added and samples were centrifuged for 10 min at $11,000 \times g$. 80 μL of the lower organic phase after extraction was transferred to a glass vial and dried in a speed-vac without heating. For transmethylation reactions, pellets were re-dissolved in 40 μL TBME (tert-Butyl methyl ether, Sigma 306975) and 20 μL TMSH (Trimethylsulfoniumhydroxid, Sigma 92732), incubated for 45 min at 50 °C and analyzed using a GC/MS-QP2010 Plus (Shimadzu®) fitted with a Zebron ZB 5MS column (Phenomenex®; 30 meter × 0.25 mm × 0.25 μm) for fatty acid methyl esters (FAME). The GC was operated with an injection temperature of 230 °C and 1 μL sample was injected with split mode (1:10). The GC temperature program started with a 1 min hold at 40 °C followed by a 6 °C/min ramp to 210 °C, a 20 °C/min ramp to 330 °C and a bake-out for 5 min at 330 °C using helium as carrier gas with constant linear velocity. The MS was operated with ion source and interface temperatures of 250 °C, a solvent cut time of 7 min and a scan range (*m/z*) of 40–700 with an event time of 0.2 s. The "GCMS solution" software (Shimadzu®) was used for data processing.

## PET scan

Positron emission tomography (PET) scan was conducted during an ITT. Insulin administration was directly followed by an intravenous injection 5 MBq (maximum volume 0.1 ml) of the radioactively labeled glucose derivative $^{18}$F-fluorodeoxyglucose ($^{18}$F-FDG; in-house production, German Cancer Research Centre Heidelberg) under isoflurane anesthesia in the PET scanner. $^{18}$F-FDG was dissolved in an isotonic buffer (adjusted to pH 7.4). Comparable to regular glucose, the radioactive tracer $^{18}$F-FDG is transported into peripheral tissues in an insulin-dependent manner. Intracellularly, it is then phosphorylated by hexokinases, similar to glucose. Phosphorylated $^{18}$F-FDG cannot be further metabolized and thus accumulates inside the cells ("metabolic trapping"). Over 60 min after the insulin injection, a dynamic PET scan was done in list mode and, thereafter, 28 time frames were reconstructed. The time activity curves were then used for a pharmacokinetic modeling using the software package PMOD (Haberkorn et al, 1997; Haberkorn et al, 2001). In addition, the last 10 min of the PET scan were reconstructed to obtain an endpoint image and the standardized uptake value (SUV) was calculated.

## Chromaffin cell preparation

Immediately after the mouse was sacrificed, both adrenal glands were excised and immersed in ice-cold Locke's solution, composed of (in mM) 154 NaCl, 5.6 KCl, 3.6 NaHCO₃, 5 HEPES, 5.6 glucose, adjusted to pH 7.3. Glands were trimmed of excess fat and connective tissue. Afterwards, the adrenal cortex was dissected from the medulla. Medullae were incubated in 1 mL of digestion solution (0.2 mg/ml L-Cystein, 100 mM CaCl₂, 50 mM EDTA, 20–25 U/ml papain (Worthington); bubbled with 95% O₂/5% CO₂ for 20 min) for 25 min at 37 °C. Subsequently, medullae were washed 2× with 1 ml of DMEM growth medium, supplemented with 0.4% PenStrep, and 1% ITSX (Themo Fisher Scientific),

followed by incubation in 1 mL of stop solution for 3–5 min at 37 °C. After washing 2× more with growth medium, tissues were triturated in 120 μL of growth medium—first with a 1000 μL, then with a 200 μL pipette. Afterwards, 20 μL each of the cell suspension were plated onto 6 glass coverslips (diameter: 18 mm) in a 12-well plate and cells were allowed to settle and adhere to the glass for 30 min (37 °C in 5% CO₂) before 1 mL of growth medium was added to each well. Cells were incubated minimum 16 h before their use in experiments.

## Ca²⁺ imaging chromaffin cells

Ca²⁺ signals were measured in mouse chromaffin cells after 2–3 days in culture (DIC). Cells were loaded with Fura-2-acetoxymethyl ester (2 μM; Fura-2AM; Life Technologies) for 30 min at 37 °C/8% CO₂ in culture medium, followed by an incubation in extracellular solution (ECS, in mM: 140 NaCl, 4 KCl, 10 HEPES, 35 Glucose, 2 CaCl₂, 1 MgCl₂, pH 7.3) for 15 min at RT to allow for the hydrolysis of the ester. Cells were immediately imaged on an inverted Zeiss AxioVert 200 microscope using a 40× (Zeiss, NA 1.3) oil objective. 340/380 nm fluorescence was excited using a Polychrome V monochromator (Till Photonics, Germany) and recorded with an Evolve EMCCD camera (Visitron, Germany) at a 16 Hz acquisition rate controlled by VisiView (Visitron, Germany). Images were processed offline using ImageJ 1.52a software and SigmaPlot 13. Images were background subtracted using the background subtraction tool. Changes in $[Ca^{2+}]_i$ were quantified within regions of interest (ROI) throughout the recording.

## Electrophysiology on chromaffin cells

Experiments were performed on mouse adrenal chromaffin cells prepared from adult (16–20 weeks old) male wild-type, *Trpc5$^{-/0}$*, *Trpc5$^{fx/0}$;Cre$^-$* mice or *Trpc5$^{fx/0}$;Cre$^+$* mice. Recordings were done at room temperature on day 2 in culture.

For PACAP-38 application (1 μM, Sigma-Aldrich), a gravity-fed multi-channel superfusion pipette was employed. TRPC1/4/5 activity was blocked by supplementing Ringer's solution (5 min pre-incubation) and the stimulating PACAP-containing Ringer's solution with HC-070 (50 nM, Med Chem Express). Carbon fiber electrodes (5 μm diameter, Amoco) manufacturing and amperometric recordings with an EPC-7 amplifier (HEKA Elektronik, electrode voltage +800 mV) were done as described previously (Bruns, 2004). Amperometric currents were filtered at 3 kHz and digitized gap-free (25 kHz). For data collection and analysis, the programs pClamp6 (Axon Instruments) and a customized event detection routine (Bruns et al, 2000) based on AutesW (NPI Electronics) were used. The analysis was restricted to events with a peak amplitude >3 pA and a total charge ranging from 10 to 1000 fC. The start of the foot signal is defined as the time point when the current amplitude exceeds two times the standard deviation of the average baseline noise. Foot signal ends at the inflection point between the slowly increasing foot signal and the more rapidly increasing spike current.

The extracellular Ringer's solution used for all electrophysiological recordings contained (in mM): 130 NaCl, 4 KCl, 2 CaCl₂, 1 MgCl₂, 30 glucose, 10 HEPES-NaOH, pH 7.3 (osmolarity adjusted to 310 mOsm). Current and voltage recordings were performed at room

temperature using the perforated patch clamp technique. Amphotericin B (dissolved in DMSO) was added to a pipette solution containing (in mM): 75 $K^+$-Glutamate, 10 HEPES, 8 NaCl, 1 $CaCl_2$, 121 glucose, 0.26 amphotericin B, osmolarity 300 mOsm, pH = 7.3 with KOH. The pipette solution was freshly made every 4 h. Thick-walled borocilicate pipettes (pipette resistance 4–5 M$\Omega$) were tip-filled with Ringer solution before back-filling with intracellular solution. After establishment of the on-cell configuration, the development of access resistance was continuously monitored, and recordings were performed on cells having a stable access resistance of less than 15 M$\Omega$. Inward currents were stimulated by applying 1 µM PACAP-38 in Ringer's solution for 120 s using a gravity-fed perfusion system. PACAP-stimulation was bracketed by superfusing cells with Ringers' solution. Cell membrane capacitance (CM) was determined during the baseline and the washout phase. Data were acquired with the Pulse software (HEKA, Lambrecht, Germany) and capacitance measurements were performed according to the Lindau–Neher technique (sine wave stimulus: 1000 Hz, 35 mV peak-to-peak amplitude, DC-holding potential −70 mV). Whole cell voltage clamp recordings were performed to measure PACAP-activated cation currents. To facilitate the isolation of TRPC5 currents, activity of voltage-gated $Na^+$ and $K^+$-channels was reduced by recording cells in Ringer's solution containing (in mM): 130 NaCl, 2 $CaCl_2$, 1 $MgCl_2$, 10 TEA-Cl, 4 CsCl, 10 HEPES, 10 Glucose, 0.1 TTX, pH 7.4 with NaOH and 320 mOsm. The intracellular solution contained (in mM): 75 $Cs^+$-Glutamate, 5 NaCl, 0.28 EGTA, 0.18 $CaCl_2$, 20 TEA-Cl$^-$, 2 $Mg^{2+}$-ATP, 0.3 $Na^+$-GTP, 90 Glucose, 10 HEPES, pH 7.4 with CsOH and 315 mOsm. After the establishment of the whole cell configuration cells were stimulated with a ramp-like voltage profile (−70 to +60, 400 ms duration) before PACAP application and during PACAP-induced current activity. Current and voltage signals were digitized at 20 kHz. Signals were analyzed with customized IgorPro routines (Wavemetrics, Lake Oswego, OR). The cumulative charge reflects the time integral of the charge transfer over the observation period.

## Intracerebral virus injection

*Trpc5$^{fx/0}$* animals were deeply anaesthetized by an intraperitoneal (i.p.) injection of a mixture of fentanyl (0.03 mg/kg body weight), medetomidine hydrochloride (1 mg/kg) and midazolam (10 mg/kg). The fur on the head was shaved off. A medial skin incision was made to expose the skull and holes (200 µm in diameter) were drilled above the regions of interests. The recombinant adeno-associated viral (rAAV) particles were injected along the following coordinates to infect bilaterally the ventrolateral medulla (VLM) areas: relative to Bregma, 6.64 mm posterior, 1.25 mm lateral and at depths of 4.35, 4.5, and 4.6 mm; 7.56 mm posterior, 1.2 mm lateral, and at depths of 4.05, 4.15, and 4.3 mm, according to the mouse brain atlas (Paxinos and Franklin, 2001).

The *Trpc5$^{fx/0}$* mice were randomly allocated to receive either rAAV2-synapsin-iCre-eGFP to selectively inactivate *Trpc5* (named *Trpc5$^{fx/0}$;VLM-Cre*) or rAAV5-synapsin-eGFP as a control virus (identifier v229 and v81, respectively; both purchased from University of Zurich viral vector facility, Switzerland; 0.05 µL over 5–10 min at each site). Sequence information for both plasmids is available at https://www.addgene.org. After the injections, the skin incision was sutured and lidocaine (10%) was applied around the incision area. The animal's anesthesia was antagonized, applying a defined mixture of atipamezole (2.5 mg/kg body weight), flumazenil (0.5 mg/kg), and naloxone (0.4 mg/kg).

For post-surgical analgesia carprofen (5 mg/kg body weight) was administered subcutaneously. Animals were left to recover in warmed cages and kept for 3 weeks to achieve optimal viral expression. At the end of experiments, viral expression was analyzed. Animals not exhibiting viral expression in the regions of interest were excluded from the study.

## Immunostaining in tissue slices

### Vibratome sections of the brain stem following stereotactic AAV injections

At the end of the ITT, after blood sampling, the AAV-injected *Trpc5$^{fx/0}$* animals were anesthetized with a mixture of ketamine (138 mg/kg body weight) and xylazine (19 mg/kg) and transcardially-perfused with ice-cold phosphate buffered saline (PBS; 0.1 M, pH 7.4), followed by 4% paraformaldehyde (PFA). The brain was removed and post-fixed overnight at 4 °C in 4% PFA before being transferred to 0.5% PFA. Coronal sections (50 µm thick) were cut with a vibratome and collected in 0.5% PFA. The free-floating sections were incubated in an antigen retrieval solution (2.94% tri-sodium citrate in distilled water, pH 8.5) at 83 °C. After cooling to room temperature (RT), the sections were incubated in 50 mM glycine in PBS followed by 0.2% triton in PBS (PBST), then washed twice with 10% normal horse serum (NHS) in PBS. The sections were incubated overnight at 4 °C with the primary antibodies (chicken anti-tyrosine hydroxylase (TH), Abcam: ab76442; rabbit anti-NeuN, Abcam: ab177487), both diluted with 10% NHS in PBST 1:1000. After washing twice with 10% NHS in PBS, the secondary antibodies were applied (goat anti-chicken Alexa 647, Thermo Fisher Scientific: A32933; donkey anti-rabbit Alexa 594, Thermo Fisher Scientific: A32754), both diluted with 10% NHS in PBS 1:700, for 1 h at RT. Following this, the brain slices were washed again with 10% NHS in PBS and incubated in Hoechst 33342 (1:10,000 in PBS; Thermo Fisher Scientific) for 10 min. The sections were subsequently washed in PBS, incubated in TRIS-HCl (10 mM), and mounted onto microscope slides with Mowiol.

Immunofluorescence was visualized and quantified with a laser-scanning confocal microscope (Leica TCS SP5 and 8) using identical illumination parameters for sections prepared within each staining trial. Images (4 µm planes) were obtained and z-stacked images were maximally projected (Leica Application Suite X) and overlaid with the corresponding anatomical atlas for detailed evaluation in the regions of interests. For analysis of the AAV transduction rate, the number of Cre-transduced TH-positive neurons in relation to the total number of TH-positive cells was determined, using the image freeware FIJI.

### Cryostat sections of Trpc5-IRES-Cre;eR26-τGFP mice

Experimentally naive *Trpc5-IRES-Cre;eR26-τGFP* mice (Schwarz et al, 2019) were anesthetized with a mixture of ketamine (138 mg/kg body weight) and xylazine (19 mg/kg) and transcardially perfused with 4% ice-cold PFA. After the post-fixation in PFA over 3 h at 4 °C, brains and adrenal glands were kept in 18% sucrose overnight, again at 4 °C, for dehydration of the tissues. Brains were then embedded in Tissue Freezing Medium® (Leica Biosystems) and stored at −80 °C until coronal sections (brain: 14 µm, adrenal gland: 10 µm) were cut at −19 °C (brain) or −15 °C (adrenal gland) on a cryostat (Leica CM3050 S) and mounted onto adhesive microscope slides (SuperFrost Plus®, Thermo Scientific), which were kept at −80 °C. To prepare the tissue slices for immunostaining, they were dried at RT and washed in PBS.

Subsequently, the tissue was simultaneously blocked and permeabilized in 10% normal donkey serum, 3% BSA, and 0.3% TX-100 (in PBS). Overnight, the slices were incubated at 4 °C with the primary antibodies (chicken anti-GFP, Thermo Fisher Scientific: A10262; rabbit anti-tyrosine hydroxylase, Millipore: ab152), diluted in PBS 1:1000. After washing in PBS with Tween (PBST), the secondary antibodies were applied (GFP-labeled anti-chicken, Jackson ImmunoResearch: 703-225-155; Cy3-labeled anti-rabbit, Jackson ImmunoResearch: 711-165-152, both raised in donkey), diluted in PBST 1:500, at RT for 2 h. Finally, the slices are stained with BisBenzimide (2 μg/mL in PBST) for 5 min before cover slipping the slides with Fluoromount-G® (SouthernBiotech). Fluorescent images were taken with the confocal microscope Leica SP5-II (10× magnifying objective) using the Leica software LAS AF. Images were processed using the image freeware FIJI.

### Detection of TRPC5 in catecholaminergic neurons of the RVLM and NTS

The immunohistochemical procedure was performed on coronal brain slices (Bregma −5.14 – −7.56 mm) from male mice (6–15 weeks) adapting a previously described method (Blum et al, 2019). The adult mice were anesthetized by injection of ketamine-HCl and xylazine (150 mg/kg and 10 mg/kg body weight, respectively) and perfused with ice-cold PBS, followed by 4% PFA in PBS. Brain samples were dissected, post-fixed for 3 h in 4% PFA at 4 °C, and cryoprotected in 30% sucrose for 2 days. Samples were then embedded in OCT compound (Tissue-Tek), and snap frozen in a bath of dry-ice/2-methylbutane. Frozen tissue was stored at −80 °C until further use. Coronal cryosections (hypothalamus/hippocampus: Bregma −1.6 – −201 mm; A1/C1 area of the rostral ventro-lateral medulla (VLM) and A2/C2 area of the nucleus tractus solitarii (NTS): Bregma −6.48 – −7.56 mm) (Bucci et al, 2017) of 18 μm were cut on a Leica CM3500 cryostat and collected on Superfrost Plus glass slides. The hypothalamus/hippocampus sections were used as controls to confirm and verify TRPC5 immunostaining in wild-type and *Trpc5⁻/⁰* mice. Sections were washed with PBS prior to heat-induced antigen retrieval under acidic conditions (1 × HIER citrate buffer pH 6.0, Zytomed Systems) for 2 min in a pressure cooker. After cooling for 15 min to RT (20 °C), sections were washed, first in distilled water and subsequently in PBS for 10 min each. Slices were then treated with 3% $H_2O_2$ for 10 min, washed with PBS, incubated in endogenous biotin-blocking kit (Invitrogen; 15 min streptavidin reagent and 15 min biotin reagent), washed with PBS and incubated in blocking buffer containing 0.3% Triton X-100 (Sigma), 4% normal goat serum (Vector Laboratories), and 2% BSA (Sigma) prepared in TBS (100 mM TrisCl, 150 mM NaCl, pH 7.5) for 90 min. All following steps were carried out at RT, incubation of primary antibodies was at 4 °C. For simultaneous detection of TH- and TRPC5-immunoreactivity, indirect immunohistochemistry (TH) and TSA amplification (TRPC5) were carried out as follows. Sections were incubated with unconjugated AffiniPure Fab fragment goat anti-mouse IgG (H + L; #115-007-003, Jackson ImmunoResearch; RRID:AB_2338476) in TBS-blocking solution for 24 h to block endogenous IgG and washed with TBS-blocking solution for 1 h, followed by incubation in blocking solution containing mouse anti-TRPC5 (1:500, monoclonal; clone N67/15, NeuroMab, Davis, CA; RRID:AB_2240979) antibody for 24 h. Tissue sections were rinsed 3× 10 min in TBS and incubated in biotinylated goat anti-mouse IgG2b antibody (1:400, #115-065-207, Jackson Immuno Research; RRID:AB_2338573) for 1 h, in streptavidin-HRP (1:100, TSA-Biotin System, Perkin Elmer) for 30 min, in biotinylated tyramid (1:100, TSA-Biotin System, Perkin Elmer) for 10 min, and in Alexa 488-

conjugated streptavidin (1:200; S-32354, Invitrogen; RRID:AB_2315383) for 90 min, with three TBS washing steps in between reagents. Then, sections were rinsed for 10 min in TBS-buffer (100 mM Tris, 150 mM NaCl, pH 7.5) and washed for 60 min in TBS-blocking solution (0.3% Triton X-100 (Sigma), 4% normal donkey serum (Vector Laboratories), 2% BSA (Sigma)). Brain sections were incubated with rabbit anti-Th (1:1000, polyclonal, #ab112, Abcam; RRID: AB_297840) diluted in blocking buffer overnight at 4 °C. Secondary antibody was: donkey anti-rabbit-Alexa555 (1:1000, #A31572, Thermo Fisher; RRID: AB_162543). Nuclei were counterstained with Hoechst 33342 (1:10,000, Invitrogen) for 12 min, washed 3× in TBS for 10 min and coverslipped with fluorescence mounting medium (DAKO). Fluorescence images were taken with a DP71 camera (Olympus) attached to a BX61 epifluorescence microscope.

## Data analysis

Data were analyzed in OriginPro, Microsoft Excel, IgorPro, Sigma-Plot13 and GraphPad Prism. Sample size was estimated prior to the experiments, based on similar experiments and estimated effect size within our research teams or throughout the scientific literature. Results are shown as mean ± s.e.m., data was collected from at least two independent biological replicates, with n representing the number of mice for the in vivo data and the number of cells for electrophysiological recordings. Statistical significance was generally determined using two-tailed unpaired Student's *t*-test, the use of a paired test is indicated in the figure legend. Normality of the datasets was assessed with the Shapiro–Wilk test. If one data set in an analysis was non-parametric distributed the Mann–Whitney U test was performed. Data of three independent samples were analyzed with one-way ANOVA. When two samples differed in two factors, independent of each other, data were analyzed using two-way ANOVA. Two-way ANOVA tests were followed by Bonferroni's post hoc analysis to test specific pair comparisons. Differences with $p < 0.05$ were considered statistically significant. Significances were depicted as *$p < 0.05$, **$p < 0.01$, and ***$p < 0.001$. Insulin tolerance tests: At any time point after 0 min, the differences in the blood glucose levels between the insulin- and saline-treated groups are statistically significant with $p < 0.001$, unless the significance level is explicitly indicated.

## Data availability

All relevant data are included in the article and/or expanded view figure, the appendix and/or the source data. Source data for the exemplary electrophysiological recordings can be accessed on the BioStudies server with Accession Number S-BSST1529. Information and reasonable requests for resources and reagents should be directed to and will be fulfilled by the corresponding author.

The source data of this paper are collected in the following database record: biostudies: S-BSST1529.

## Peer review information

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

## Acknowledgements

We thank Nadine Gehrig, Manuela Ritzal, Irina Kupin, Hans-Peter Gensheimer and the entire team of the Interfakultäre Biomedizinische Forschungseinrichtung (IBF) from Heidelberg University for expert technical assistance as well as LetPub for data visualization. DBH-cre mice were provided by Dr. Günther Schütz, DKFZ, Germany. We thank the Metabolomics Core Technology Platform of the Centre for Organismal Studies (COS) at the University of Heidelberg. This work was supported by Young Investigator Grant of the Saarland University (YS), the Collaborative Research Centers of the DFG (German Research Foundation - Deutsche Forschungsgemeinschaft): TR-SFB 152 (MF, DB, YS, TLZ, FZ, UB), CRC 1118 (MF, RK, NA, PN, RH), CRC 1550 (MF, RTL), and CRC 894 (DB, TLZ, FZ, MF).

## Author contributions

Jenny Bröker-Lai: Conceptualization; Data curation; Formal analysis; Investigation; Visualization; Methodology; Writing—original draft; Writing—review and editing. José Rego Terol: Formal analysis; Investigation; Methodology. Christin Richter: Investigation; Methodology. Ilka Mathar: Conceptualization; Investigation. Angela Wirth: Conceptualization; Resources; Investigation; Methodology; Project administration. Stefan Kopf: Resources; Investigation; Methodology. Ana Moreno-Pérez: Investigation; Methodology. Michael Büttner: Formal analysis; Investigation; Methodology. Linette Liqi Tan: Investigation; Methodology. Mazen Makke: Formal analysis; Investigation; Methodology. Gernot Poschet: Conceptualization; Formal analysis; Investigation; Methodology; Writing—review and editing. Julia Hermann: Formal analysis; Investigation; Methodology. Volodymyr Tsvilovskyy: Conceptualization. Uwe Haberkorn: Formal analysis; Investigation. Philipp Wartenberg: Investigation; Methodology. Sebastian Susperreguy: Investigation; Resources. Michael Berlin: Formal analysis; Visualization. Roger Ottenheijm: Formal analysis; Visualization; Writing—review and editing. Koenraad Philippaert: Data curation; Formal analysis; Visualization; Project administration; Writing—review and editing. Moya Wu: Investigation; Methodology. Tobias Wiedemann: Formal analysis; Investigation; Methodology. Stephan Herzig: Supervision. Anouar Belkacemi: Conceptualization; Investigation; Visualization; Methodology. Rebecca T Levinson: Data curation; Formal analysis. Nitin Agarwal: Supervision; Investigation; Methodology. Juan E Camacho Londono: Formal analysis; Investigation; Methodology. Bert Klebl: Resources; Formal analysis. Klaus Dinkel: Resources; Formal analysis. Frank Zufall: Conceptualization; Supervision; Writing—review and editing. Peter Nussbaumer: Conceptualization; Resources; Formal analysis; Writing—review and editing. Ulrich Boehm: Resources; Supervision. Rüdiger Hell: Resources; Supervision. Peter Nawroth: Resources; Supervision; Investigation. Lutz Birnbaumer: Resources. Trese Leinders-Zufall: Conceptualization; Supervision; Writing—review and editing. Rohini Kuner: Supervision; Investigation; Methodology. Markus Zorn: Supervision; Investigation. Dieter Bruns: Conceptualization; Resources; Supervision; Funding acquisition; Investigation; Writing—original draft; Project administration; Writing—review and editing. Yvonne Schwarz: Conceptualization; Supervision; Funding acquisition; Investigation; Visualization; Methodology; Writing—original draft; Project administration; Writing—review and editing. Marc Freichel: Conceptualization; Resources; Data curation; Formal analysis; Supervision; Funding acquisition; Investigation; Writing—original draft; Project administration; Writing—review and editing.

Source data underlying figure panels in this paper may have individual authorship assigned. Where available, figure panel/source data authorship is listed in the following database record: biostudies: S-BSST1529.

## Funding

## Disclosure and competing interests statement

The authors declare no competing interests.

