## [Peer Review File · The EMBO Journal]

TRPC5 controls the adrenaline-mediated counter regulation of hypoglycemia

Jenny Bröker-Lai, José Rego Terol, Christin Richter, Ilka Mathar, Angela Wirth, Stefan Kopf, Ana Moreno-Pérez, Michael Buettner, Linette Liqi Tan, Mazen Makke, Gernot Poschet, Julia Hermann, Volodymyr Tsvilovskyy, Uwe Haberkorn, Philipp Wartenberg, Michael Berlin, Roger Ottenheim, Koenraad Philippaert, Moya Wu, Tobias Wiedemann, Stephan Herzig, Anouar Belkacemi, Rebecca T Levinson, Nitin Agarwal, Juan E. Camacho Londono, Bert Klebl, Klaus Dinkel, Frank Zufall, Peter Nussbaumer, Ulrich Boehm, Rüdiger Hell, Peter Nawroth, Lutz Birnbaumer, Trese Leinders-Zufall, Rohini Kuner, Markus Zorn, Dieter Bruns, Yvonne Schwarz, and Marc Freichel

Corresponding authors: Marc Freichel (marc.freichel@pharma.uni-heidelberg.de) , Yvonne Schwarz (yvonne.schwarz@uks.eu)

Review Timeline:

Submission Date:	14th Jun 24
Editorial Decision:	29th Jun 24
Revision Received:	10th Jul 24
Accepted:	12th Aug 24

Editor: Kelly Anderson

Transaction Report:

This manuscript transferred to The EMBO Journal following peer review at another journal. The peer review comments and authors' responses were made available as agreed with the authors and the other journal, and were taken into account for the decision process at The EMBO Journal.

Point-by-point reply to the Reviewers' comments:

Reviewer #1 (Remarks to the Author):

This manuscript by Marc Freichel's group is an excellent paper that convincingly reports data at the cellular and molecular level that TRPC5 channels control catecholamine (CA, adrenaline) release from adrenal chromaffin cells (CCs) and in this way can regulate body hypoglycemia via PLC.

Using various TRPC compound KO mouse lines, in particular the TRPC5^{-/-} mouse, the authors outline the failure of sympathetic counter regulation to the loss of TRPC5 channels in adrenal chromaffin cells (CCs), resulting in reduced blood plasma adrenaline rise.

Using GFP-immunofluorescence, the authors first show clearly that the medulla of adrenal gland slices express TRPC5 channels and that these TRPC5 channel subtypes regulate calcium entry and CA release induced by PACAP and muscarine (both being the neurotransmitters released from the splanchnic nerves innervating adrenal CCs). The authors then prove that PACAP (PAC1) and muscarine (M1) induce CA release through the activation of TRPC5 channels and that the activation is mediated by the PLC pathway.

Using current-clamp recordings, the authors show clearly that PACAP induces a significant CCs depolarization, which does not cause action potential (AP) firing but is mediated by TRPC5 channels. This rules out the possibility that CA release is likely induced by high-threshold voltage-gated Ca²⁺ channels open during the APs. An interesting argument that needs to be clarified in future studies.

We thank the reviewer for the positive judgment of our work and her/his enthusiasm for the paper.

Minor Criticism:

In my view, the authors stress too much the role of PACAP on TRPC5 channel activation but, interestingly they also report the clear involvement of muscarinic receptors on this action. A mechanism that will be certainly better clarified in following studies.

We thank the reviewer for their help to improve the presentation and interpretation of our data. Our results delineate a so far unrecognized signaling pathway of how muscarinic and PAC1 receptors at the adrenomedullary synapse trigger sustained adrenaline secretion through the activation of TRPC5 channels. To accommodate the reviewer's suggestion, we have now added the muscarinic receptor into our graphical abstract.

Below are indicated few minor errors:

1 – p.10 1.313 and 314: the indications to the GFP and TH fluorescence should be changed.

The requested changes have been made in the revised manuscript

2 –p. 12 1.376: change D-F in D-E

The requested changes have been made in the revised manuscript

Reviewer #2 (Remarks to the Author):

The manuscript by Broker-Lai and colleagues is a compelling basic science study revealing a vital role of TRPC 5 channels in maintaining adrenaline secretion in insulin-evoked hypoglycemic states. The authors initially showed that male mice lacking TRPC1/TRPC4/TRPC5/TRPC6 channels develop aggravated hypoglycemia following insulin treatment, and then they demonstrated that genetic ablation of *Trpc5* was sufficient to recapitulate the hypoglycemic state. Furthermore, the authors showed that treatment with adrenaline mitigated the aggravated hypoglycemia in the *Trpc5* KO mice. They then used TRPC 5 channel agonists and antagonists to establish a clear role of these channels in regulating adrenaline secretion in hypoglycemic states. Using conditional KO mouse models, the authors pinpoint the locus of this regulation to the adrenal chromaffin cells. Indeed, the loss of TRPC5 channel activity in chromaffin cells impaired the PACAP- and muscarinic-mediated (both released by sympathetic nerves) secretion of adrenalin. Using multiple cellular electrophysiological techniques including calcium imaging and perforated patch recordings, the authors convincingly demonstrated that the PACAP and muscarinic excitation of chromaffin cells is dependent on the activation of TRPC5 channels. The excitation is lost with *Trpc5* mutagenesis or antagonism of TRPC5 channels. The authors established that the PACAP signaling is dependent on the phospholipase C pathway using pharmacological tools.

We thank the reviewer for these positive remarks on our paper.

Major Criticism:

1) The only biophysical measurement missing is a current/voltage plot showing the double rectification that is the hallmark of TRPC5 channel activation (e.g., Blair et al., J Gen Physiol, 2009).

Ad 1) To address the reviewer's request, we have now extended the biophysical characterization of PACAP evoked currents in primary chromaffin cells of wildtype and *Trpc5*⁻⁰ mice using whole-cell patch clamp experiments (Fig. Q&A 2 below and Fig. S7A, B in the revised manuscript). In the presence of TTX, TEA and Cs⁺, we were able to isolate PACAP induced TRPC5 currents. Clear inward and outward currents were observed in response to a voltage ramp protocol in wt cells with a reversal potential of -11 mV indicative of a nonselective cation conductance. Furthermore, in cells isolated from *Trpc5*⁻⁰ mice, the non-selective cation conductance was abolished, indicating its dependence on TRPC5 activity. Moreover, the properties of the observed PACAP induced TRPC5 currents agree well with those of previous published TRPC5-dependent currents in HEK293 cells (Blair et al., J Gen Physiol, 2009) and in neurons from the lateral nucleus of the amygdala (Ricchio et al., 2009, PMID: 19450521). We have included this data as new Supplementary figure S7 A, B and in the text (p13, line 380-390) of the revised manuscript.

Figure Q&A 2: (A, B) Exemplary current-voltage relationship (IV) of wt (A) and *Trpc5*^{-/-} chromaffin cells (B) before (black) and during (red) PACAP-induced burst-like current activity (voltage ramp from -70 to +60 mV, 400 ms duration). wt cells show bell-shaped Ca²⁺-current activity under baseline conditions (black) and respond with additional strong inward and outward currents during PACAP-application (red). PACAP-induced currents (blue) of wt cells exhibit a reversal potential (V_{rev}) around -11 mV. PACAP-induced inward and outward currents depend on TRPC5 channel activity, as no comparable currents were detected in *Trpc5*^{-/-} cells.

This figure is represented in supplemental figure S7 A, B in the revised manuscript.

2) Overall, the mouse experiments are well-executed and convincingly demonstrate the dependence of adrenaline secretion from chromaffin cells on TRPC5 channel activation. However, the metabolic parallels between TRPC5 deficiency in mice and the human conditions of hypoglycemia-associated autonomic failure (HAAF) are tenuous. The authors attempted to identify commonalities between the TRPC5 deficiency in mice and HAAF in diabetic patients. Although they found that insulin-induced hypoglycemia reduced the concentrations of numerous amino acids in conditional *Trpc5* KO mice, they only found differences in the concentrations of three amino acids in HAAF patients versus non-HAAF patients under hypoglycemic conditions. Taurine was the only amino acid that was common to both HAAF patients and *Trpc5*^{fx/0}; DBH-Cre⁺ mice (i.e., Figure 9). However, these measurements in humans were essentially “one-off” experiments (a single sample at one time point) and do not take into account the complexity of these metabolic pathways. The human experiments are by far the weakest component of otherwise solid and well-executed mouse experiments.

Ad 2) We agree with the reviewer’s point and therefore in the revised manuscript only draw the attention to similarities regarding the reduced adrenaline secretion and plasma taurine levels in *Trpc5*^{-/-} mice and HAAF patients under hypoglycemic conditions. Yet, HAAF is a syndrome, characterised by defects in several physiological defense mechanisms against hypoglycemia rather than on the disruption of a singular biochemical reaction (PMID: 23883381). Therefore, a plurality of underlying mechanisms could cause HAAF, one of them might be altered TRPC5 activity. In any case, our experiments do not allow us to conclude that TRPC5 deficiency is the sole cause of the HAAF syndrome.

Minor Criticism:

1. Figure 7 E and F are mislabeled. The stats in the legend are incorrect for the Figures.

Thank you for the remark. We have corrected this in the revised manuscript.

2. Figure 4G should be enlarged since the near perfect co-localization is important to see.

We have enlarged these panels in the revised manuscript.

3a. For Figures 6 and 7, the authors need to state how cumulative charge was determined in the Methods for the non-electrophysiologist readers.

Ad 3a) The cumulative charge reflects the time integral of the charge transfer over the observation period. This information has been added to the Method section of the revised manuscript.

3b. Also, the label for the y-axis label in Figure 6G is missing.

Ad 3b) We have included the label for the y-axis of figure 6G in the revised manuscript.

4. The legend for Figure S4 is incomplete.

Ad 4) We have completed the legend of Figure S4, which is now Figure S5 in the revised manuscript.

5. In the legend for Figure S3B, $Trpc5^{+/+}$ should be $Trpc5^{+/0}$?

Ad 5) We thank the reviewer for pointing this out and have corrected it in the revised manuscript.

6. Also, Figure S3B should be enlarged since it represents the “transduction efficiency” in RVLN neurons. Perhaps Figure S3A could be reduced since it only depicts the neurosurgical approach.

Ad 6) We have increased the size of figure S3B in the revised manuscript.

Reviewer #3 (Remarks to the Author):

General Comments

The authors initially observed that TRPC1/4/5/6 deletion exacerbated hypoglycemia following insulin treatment. Finally, they demonstrated TRPC5 expression in chromaffin cells of the adrenal medulla to cell-autonomously play a role in secretion of adrenalin from these cells. Thus, they elucidated a mechanism underlying an artificial phenomenon, i.e., “TRPC1/4/5/6 deletion-induced hypoglycemic aggravation”, but not the (patho)physiological mechanism underlying “hypoglycemia-associated autonomic failure (HAAF)”. In addition, despite performing many in vivo experiments, they showed only the intracellular mechanism underlying adrenalin secretion by chromaffin cells. Given the contexts, this reviewer’s enthusiasms for this manuscript were attenuated.

We thank the reviewer for their detailed evaluation of the manuscript.

Specific Comments

1. The authors initially observed that TRPC1/4/5/6 deletion exacerbated hypoglycemia in response to insulin treatment. Then, they examined the effects of TRPC5 single KO on hypoglycemia. However, before producing TRPC1/4/5/6 KO mice, TRPC5 single KO mice must have been established. Therefore, the scientific reason for first examining TRPC1/4/5/6 mice was unclear. In addition, they did not compare the phenotypes between TRPC1/4/5/6 KO (or TRPC1/4/5 KO) and TRPC5 single KO mice in detail. Therefore, it remains unclear whether TRPC proteins other than TRPC5 contribute, to any extent if at all, to the worsened hypoglycemia phenotype.

Ad 1) Starting point of our studies was the observation of a severe mortality of STZ-induced diabetes in *Trpc1/4/5/6^{-/-}* mice during a study looking for longterm complications of chronic hyperglycemia in the retina (Figure 1; Sachdeva et. al, Molecular Metabolism, 2018, PMID: 29373286) unless we reduced the insulin dose. Given that TRPC channels form heterotetramers and can act in a functional redundant manner, it is often strategically necessary to employ compound knock-out models to unravel a clear phenotype (e.g. Camacho-Londono et al. Eur H Journal, 2015, PMID: 26069213; Bröker-Lai et al. EMBO Journal 2017, PMID: 28790178; Kollwe et al., Neuron 2022, PMID: 36257322). We did not repeat and compare the survival experiment depicted in Figure 1B, i.e. analysis of survival of *Trpc5⁻⁰* vs WT control mice following application of long- acting insulin glargine (1-4 Units depending on the glucose level after induction of diabetes)...

[REDACTED: Authors' response with unpublished data.]

Therefore, the data displayed in Figure 1B of the manuscript is only available for *Trpc1/4/5/6* knockout mice. We included this data set in the manuscript as these are key observations that prompted us to perform insulin tolerance tests without prior induction of chronic hyperglycemia (STZ) and led us to our central hypothesis.

[REDACTED: Authors' response with unpublished data.]

[REDACTED: Authors' response with unpublished data.]

As can be seen in the figure Q&A 3A below, the phenotype of *Trpc1/4/5/6*^{-/-} quadruple KO was equally pronounced in *Trpc5* single KO (Table 1: no significant difference is observed between *Trpc1/4/5/6*^{-/-} and *Trpc5*^{-/0} mice)... *[REDACTED: Author's response with unpublished data.]* Therefore, we continued the study as presented in the manuscript and tried to identify the responsible cell type and the molecular TRPC5-mediated mechanism in this cell type using the global and cell-specific *Trpc5*-KO mouse lines.

[REDACTED: Authors' response with unpublished data.]

Furthermore, additional immunohistochemical stainings of the adrenal medulla revealed that TRPC5 is expressed in virtually all chromaffin cells ... *[REDACTED: Authors' response with unpublished data.]*

[REDACTED: Figure for reviewers removed]

Time	Trpc1/4/5/6 ^{-/-} vs. control	Trpc5 ⁻⁰ vs. control	Trpc1/4/5/6 ^{-/-} vs. Trpc5 ⁻⁰
30 min	* p = 0,029	** p = 0,0014	n.s. p = 0,65
45 min	** p = 0,0013	** p = 0,0033	n.s. p = 0,82
60 min	* p = 0,012	** p = 0,0047	n.s. p = 0,99

[REDACTED: Authors' response with unpublished data.]

Table 1: p-values of the pairwise t-test of the blood glucose values 30, 45 and 60 min after the injection of insulin during an ITT of different *Trpc* knockout mice (from the data in figure Q&A 3 A). We observe that 1) the *Trpc1/4/5/6*^{-/-} as well as the *Trpc5*⁻⁰ mice show significantly reduced glucose values compared to their controls at 30, 45 and 60 min after insulin injection. Therefore, *Trpc1/4/5/6*^{-/-} and *Trpc5*⁻⁰ mice show a similar phenotype. 2) There is no significant difference in the blood glucose values between *Trpc1/4/5/6*^{-/-} and *Trpc5*⁻⁰ mice.

[REDACTED: Authors' response with unpublished data.]

2. Following the blood glucose measurements conducted through 60 min after insulin injection (Fig. 2C), the authors measured blood glucose levels until 150 min after insulin injection (Fig. 3A). The data in Fig. 3A include those in Fig. 2C. Adrenalin measurements were likewise conducted. The authors should delete the overlapping data.

Ad 2) We have reworked the corresponding figures and have omitted double representation in the revised manuscript. From figure 2 in question, we moved A, B, E and F to supplemental figure 1 A, B, D and E and we retained the information from panels C and D in figure 2 in the revised manuscript. This ensures a more stringent flow of the text and figures (and supplemental figures).

3. In the adrenalin rescue experiments, the authors did not measure blood adrenalin levels. To verify whether or not this supplementation procedure is physiologically appropriate, they should show the time course of adrenalin levels after adrenalin supplementation. In particular, 20 minutes after adrenalin administration, adrenalin supplementation induced higher blood glucose in TRPC5 KO than in wild-type mice, which may have contributed to producing the statistically significant difference in blood glucose levels between the saline and adrenalin groups of TRPC5 KO mice. In addition, 30 minutes after adrenalin administration, the differences in blood glucose levels between the saline and adrenalin groups were similar in wild-type and KO mice. Thus, the adrenalin rescue experiments appear to be inappropriate for showing the contribution of adrenalin reduction to worsening of hypoglycemia in TRPC5 KO mice.

1) We would like to point out that the blood glucose levels 20 min after adrenalin injection between *Trpc5*⁻⁰ and WT were **not statistically significant** different. Thus, we respectfully disagree with the statement that blood glucose values are higher in the *Trpc5*⁻⁰ adrenalin-treated group, based on the data we have. To address this issue further we performed a post hoc power analysis at that time point which reveals that this experiment is not underpowered with a power of 0.8 for $\alpha = 0.05$

given the means of (WT: 90.875 mg/dL, KO: 108.333 mg/dL and the largest SD 12.2). Within the genotypes, however, the blood glucose 20 min after adrenaline and saline treatment differs for the *Trpc5*⁻⁰ but not for WT mice (Fig. 3B). We can only speculate about the reason for the apparently (but not significantly) higher levels in the *Trpc5*⁻⁰ adrenaline group: it may well be that the *Trpc5*⁻⁰ mice, which have a disturbed adrenaline-release mechanism, respond with a higher sensitivity to the adrenalin injection than WT mice.

2) Importantly, adrenaline injections do rescue hypoglycemia in *Trpc5* KO animals. As illustrated in Fig. 2 D adrenaline injections have a significant effect on plasma glucose levels in *Trpc5* KO mice (20 min time point). Thus, this treatment eliminates the blood glucose concentration difference between *Trpc5* KO and wild-type controls even up to the time point 30 minutes after insulin injection, a time point at which - without the epinephrine treatment - there is a significant difference in blood glucose concentration between wild-type and *Trpc5* KO mice (Figure 2D, Figure 2A). Overall, the results indicate that exogenous administered adrenaline significantly attenuates insulin-induced hypoglycemia in *Trpc5*⁻⁰ mice. They strongly corroborate our conclusion that the lack of adrenaline rise is responsible for defective glucose counter-regulation.

3) Regarding adrenaline pharmacokinetics: Due to the fact that the plasma half-life time of adrenaline is very short (ca. 1-2 min^{ref*}) collecting reproducible data on blood adrenalin levels would come with practical limitations and significant challenges. It would be experimentally impossible to determine a time series of circulating adrenaline levels after adrenaline injection. Furthermore, it will be similarly difficult to record the corresponding time course of changes in blood glucose which follow with a certain time delay, as they are first triggered by the reaction of adrenaline on hepatocytes (gluconeogenesis, glycogenolysis). In our opinion, such measurements and the gathered data, while potentially interesting for other mechanisms, will not provide any new information which strengthen our conclusions in the manuscript. Therefore, such animal experiments will be hard to get approved by the local ethics committee.

ref*. Lip G, and Hall J. Comprehensive Hypertension. Mosby. 2007;1

4. The authors concluded that neither glucocorticoids nor glucagon were responsible for exacerbation of insulin-induced hypoglycemia in TRPC5-deficient mice, based on the findings indicating that these hormone levels were similar in wild-type and TRPC5-deficient mice. However, as shown in Figs. 2C and 3A, blood glucose levels were significantly lower in TRPC5-deficient than in wild-type mice. Therefore, if the secretions of these hormones were not impaired, blood levels of these hormones should have been higher in TRPC5-deficient mice. In addition, for an unexplained reason(s), the cortisol tolerance tests were performed in TRPC1/4/5/6 KO mice (not in TRPC5-deficient mice). To meaningfully compare the secretory capacities of these hormones, the authors should perform the clamp procedure to produce similar blood concentrations of glucose and insulin in wild-type and TRPC5-deficient mice and then compare the levels of these hormones.

To address the reviewers concern, we measured the rise in glucagon and corticosterone at defined glucose levels, respectively. For this, we used the mouse line with *Trpc5* deletion solely in catecholaminergic cells (*Trpc5*^{fx/0};DBH-Cre). In the revised paper, we show that *Trpc5*^{fx/0};DBH-Cre⁺ and *Trpc5*^{fx/0};DBH-Cre⁻ control mice show no difference in levels of glucagon and corticosterone when clamped in a steady state hypoglycemic condition (at 50 mg/dL glucose), (Figure Q&A 4 below and supplemental figure S3 in the revised manuscript). These results suggest that the release

of either hormone triggered by hypoglycemia into the blood is not impaired in our mouse model with deletion of TRPC5 in catecholaminergic cells, and it renders the possibility unlikely that either alterations in the rise of plasma glucagon or corticosterone are responsible for the observed phenotype. The results are in line with our concept that loss of adrenaline secretion is responsible for the aggravated hypoglycemia in *Trpc5*-deficient mice.

Another important observation in these experiments is the increased glucose infusion rate in the *Trpc5^{flv0};DBH-Cre⁺* mice compared to their controls, supporting the central hypothesis that TRPC5 is essential for the autonomic counterregulation to hypoglycemia. These results have now been added to the manuscript.

The reason for the use of quadruple knockout *Trpc1/4/5/6^{-/-}* has been addressed above (question 1). As the *Trpc1/4/5/6^{-/-}* mice do not exhibit any changes in the cortisol tolerance test, repeating this experiment in single knockout mouse lines cannot be justified to our local animal ethics committee.

We show that there is no compensatory upregulation of either of these hormonal pathways (glucagon and corticosterone) after deletion of *Trpc5*. Moreover, it is known and we show that the cortisol-induced glycemic increase in mice is rather limited (Fig. S3E). Likewise, the glucagon response is not altered between WT and *Trpc5* KO.

Taken together, the additional results from the hypoglycemic clamp study support our concept of failure of sympathetic-adrenergic counterregulation in *Trpc5*-deficient mice (glucose infusion rate) which cannot be explained by alterations in glucagon or glucocorticoid plasma levels or their action.

Figure Q&A4: The interplay between glucagon and cortisol is not altered in TRPC5 deficient mice.

(A) Schematic representation of the hyperinsulinemic hypoglycemic clamp experiments with indication of the timepoints of blood sampling. The first sample, t₀, was taken before the insulin infusion started, the second sample was taken during steady state 50 mg/dL blood glucose levels. (B) The glucose infusion rate during steady state conditions. (C) The plasma glucagon concentration before and during hypoglycemia. (D) The plasma corticosterone levels before and during hypoglycemia (50 mg/dL blood glucose) n = 9-11 per group.

This figure is represented as supplemental figure S3 A-D in the revised manuscript.

5. The authors showed that TRPC5-deficiency in adrenal chromaffin cells impaired neuron-stimulated intracellular calcium elevation and catecholamine secretion stimulated by splanchnic nerve activation. These observations lead the reviewer to speculate that this mechanism is involved in a variety of stress responses, which are not specific for hypoglycemic reactions. However, the authors stressed the importance of glycemc control alone. They should examine the role of this mechanism in responses to other stress factors.

We apologize for this misunderstanding. Although we focus in our study on hypoglycemia-evoked stress, we never state that TRPC5-mediated adrenalin secretion occurs exclusively under hypoglycemic stress conditions. To follow up on this and to address the reviewer's question regarding other stress factors, we have now included two independent stress models in *Trpc5*-deficient mice into the manuscript. In these new experiments, stress is either evoked by LPS injection or histamine injection. We show that the rise of plasma adrenalin levels induced by LPS- (figure Q&A 5A and supplemental figure S2A) and histamine-treatment (figure Q&A 5B and supplemental figure S2B), respectively, are reduced in *Trcp5*-deficient mice.

Taken together, these new results show that multiple stress models involving sympathetic-adrenergic activation rely on TRPC5-mediated adrenalin secretion, and that this pathway is not specific for (insulin-induced) hypoglycemia. We have made this more clear in the revised manuscript in the discussion on page 17 lines 510-516.

6. As noted in the title, the abstract and the discussion section, the authors concluded that TRPC5 controls counter-regulation of hypoglycemia. However, the findings presented in this manuscript ultimately suggest that TRPC5 is a molecule which constitutes a basic mechanism responsible for adrenal secretion by chromaffin cells, but not the regulatory mechanism which functions in response to hypoglycemia. In particular, the authors wanted to link this mechanism with HAAF, but HAAF is considered **to be caused by impaired sensing of low glucose levels**. Do the authors actually speculate that TRPC5 is linked with the development of HAAF? The contexts, as described throughout this manuscript, do not seem to be supported by the experimental results.

Ad 6) It is important to note that our data demonstrate that TRPC5 is instrumental for sustained adrenal secretion from chromaffin cells, which represents a crucial step in the counter-regulatory response to hypoglycemia. HAAF patients are characterized by a reduced sympathoadrenal response that causes apparent hypoglycemia unawareness. A defect in either the glucose sensing or the adrenal response thereof can lead to the symptoms observed in HAAF patients and both lead to reduced hypoglycemia-induced adrenalin plasma levels (PMID: 23883381 Cryer, NEJM). Yet, the mechanisms how and why adrenaline secretion is impaired in HAAF patients is not well understood at the molecular level. Indeed, our results show a very striking parallel in sympathoadrenal failure after hypoglycemia in *Trpc5* KO mice and in HAAF patients. Yet, our results do not allow for the conclusion that the TRPC5-dependent activation of the sympathoadrenal response represents the pathomechanism underlying HAAF.

We have adapted the interpretation of our results and conclusions, especially in the context of HAAF's pathophysiology accordingly in the manuscript. (page 20, line 640 - 640)

7. The authors selected taurine which is presented at lower concentrations commonly in TRPC5-deficient mice and HAAF subjects. To exclude the possibility that this phenomenon is merely a coincidence, however, the authors need to examine the mechanism(s) commonly underlying taurine reduction in TRPC5-deficient mice and HAAF subjects. Nevertheless, based on this finding alone, the authors proposed that taurine might serve as a new diagnostic tool for HAAF patients. It is not clear to this reviewer how and under what circumstances measuring taurine levels would actually be beneficial for HAAF patients.

Ad 7) We agree with the reviewer that our data cannot rigorously exclude the possibility that the reduced taurine levels in *Trpc5* KO mice and HAAF patients are merely a coincidence. As there are no specific diagnostic tools available to predict the susceptibility for a HAAF episode, patient management is limited to the prevention of hypoglycemia. In the framework of a comprehensive metabolite analysis, we observed a specific decrease in taurine levels under hypoglycemia as another common feature between *Trpc5* KO mice and HAAF patients. Thus, plasma taurine levels, which can be determined by routine procedures in clinical labs, might be a useful diagnostic tool to identify HAAF. However, further studies with independent and larger patient cohorts are required to study whether the decrease in taurine plasma levels under hypoglycemia could indeed serve as a diagnostic marker with sufficient sensitivity and specificity in HAAF patients or for certain HAAF subgroups. To accommodate his/her suggestion, we have adapted the interpretation of our results and conclusions, also in the context of HAAF's pathophysiology accordingly in the manuscript.

In this context it stands to reason, that the underlying mechanisms leading to changes in taurine plasma levels are very complex. Taurine is one of the most abundant free amino acids in different organs throughout the body and taurine metabolism is influenced by many metabolic pathways (PMID: 23170060). Taurine is mainly produced in the liver and central nervous system (PMID: 19094361), but taurine biosynthesis is equally found in the kidney and pancreas (PMID: 23170060). Plasma taurine levels may also be elevated from dietary intake. A reduction in plasma taurine levels can be a consequence of reduced taurine production or increased systemic consumption. In addition, taurine can be used as an energy source during severe hypoglycemic conditions and is cleared faster in the body (PMID: 10969839).

8. ITT, is usually used for examining insulin sensitivity using doses of insulin within a range that does not lead to hypoglycemia. Using the term ITT may thus cause readers confusion.

We understand the remark, and we are aware of the ambiguous definition of ITT throughout the scientific literature, one main difference is its use as a diagnostic clinical tool for patients and the other is in a research setting for lab animals. The editorial guidelines for Nature Metabolism manuscripts describing ITTs fit our methodology (Virtue et al, Nature Metabolism, 2021 PMID: 34117483), insulin dose and glucose concentration ranges. Whether ITTs in experimental procedures regularly lead to hypoglycemia, and which glucose level is defined as hypoglycemia varies. Gasco et al. (Neuroendocrinology 2021, PMID 33406519) define that an ITT has to go below 40 mg/dL, while Cozar-Castellano et al. mention in ‘animal models of Diabetes’ (springer protocols, https://doi.org/10.1007/978-1-0716-0385-7_15) that glucose levels should drop at least 50%, but not below 36 mg/dL.

During our ITT, we are using an insulin dose that does not lead to hypoglycemia (by these definitions). In our WT mice, all blood glucose values stayed above 40 mg/dL (Figures 1C, 2A, 2D, 3B, 4A, 4D, S1A, S1C and S1D). In our *Trpc5*-deficient animal models, part of the observed phenotype is indeed a severe hypoglycemia and failed counter-regulation during a normal ITT. Therefore, we think that the use of the terminology “Insulin Tolerance Test” in this context is justified.

Dear Marc,

Congratulations on an excellent revision, we are happy to move forward with publication of your work pending a minor revision that (non-experimentally) addresses the final concerns raised by the referees.

When you submit your revised version, please also take care of the following editorial items and add this also to your point-by-point response:

1. Please provide an author checklist which you can find on our website.
2. Please reduce the number of keywords to 5.
3. Please review our author guidelines and update the format of the Data Availability section, and also move this section to the end of the Methods.
4. Please add all funding information with project numbers if available to our online eJP system.
5. Please remove the author contribution section from the main manuscript.
6. Please review our new policy on conflict of interests on the EMBO author guide website and update the title of this section to: Disclosure and competing interests statement.
7. Please correct the references format to EMBO style and remove dois for published papers
8. For the appendix, please add a table of contents (with page numbers), correct nomenclature is "Appendix Figure S1" etc.
9. We include a synopsis of the paper (see <http://emboj.embopress.org/>). Please provide me with a general summary statement and 3-5 bullet points that capture the key findings of the paper.
10. We also need a summary figure for the synopsis. The size should be 550 wide by 200-440 high (pixels). You can also use something from the figures if that is easier.
11. All figures should be referenced in the main manuscript, please add a callout for Fig2H, Fig7H, and Figure S10.
12. Please remove the figures and supplemental figures and legends from the main manuscript
13. Please note that the legends for figures 5b-g is not provided in the sequential manner. This needs to be rectified.
14. Please note that the figure 7f is mislabeled as figure 7e in the manuscript. This needs to be rectified.
15. Please note that the figure 7e is not labelled in the manuscript. This needs to be rectified.
16. Please define the annotated p values ***/**/* as well as provide the exact p-values for the same in the legend of figure 4a-f; 5n; 9a-b; as appropriate.
17. Please note that the exact p values are not provided in the legends of figures 1c-d, h; 2a-h; 3d; 5b, d, f, k, q; 6g; 8c, supplementary figures 1a-e; 2a-b; 3b-c, e-f; 7e; 8a-c.
18. Please indicate the statistical test used for data analysis in the legends of figures 2a-h; 4a-f; 9a-b, supplementary figures 8a-c.
19. Please note that in figures 7d-e; there is a mismatch between the annotated p values in the figure legend and the annotated p values in the figure file that should be corrected.
20. Please note that information related to n is missing in the legends of figures 6e-g; 7e, supplementary figures 3c; 4b; 5a-b, d-e; 6a-b; 7e.
21. Please note that the error bars are not defined in the legends of figures 1c-h; 2a-h; 3b-d; 4a-f; 5b, d, f, h, k, n, q; 6e-g; 7d-e; 8b-c; 9a-b, supplementary figures 1a-e; 2a-b; 3b-f; 4b-d; 5a-b, d-e; 6a-b; 7e; 8a-c; 9a-c.
22. Please note that scale bar and its definition are missing for figure 4g, supplementary figure 4b.

Thank you for the opportunity to consider your work for publication, I look forward to your revision.

Warm wishes,
Kelly

Kelly M Anderson, PhD
Editor, The EMBO Journal
k.anderson@embojournal.org

Further information is available in our Guide For Authors: <https://www.embopress.org/page/journal/14602075/>

authorguide

Point-by-point reply to the Reviewers' comments:

Reviewer #1 (Remarks to the Author):

The authors answered exhaustively to my criticisms. They made very clear in the graphical abstract, text and figures that the long lasting catecholamine release from adrenal chromaffin cells (CCs) induced to counterbalance hypoglycemia is regulated by muscarinic and PACAP receptors through the activation of TRPC5 channels, highly expressed in CCs.

In my view, the manuscript opens very interesting lines of research to understand still unknown molecular mechanisms at the basis of chromaffin cell excitability and catecholamine secretion. I do not have any further comments on this excellent manuscript.

We thank the reviewer for their careful evaluation of the manuscript and their kind words of appreciation on our work.

Reviewer #2 (Remarks to the Author):

The authors have addressed all of my comments. The manuscript is a very thorough and compelling basic science study. My only reservation is the tenuous link of taurine as a common metabolic signature between TRPC5-deficient mice and HAAF patients ("By comparing metabolites in the plasma of TRPC5-deficient mice and HAAF patients, we identified reduced taurine plasma levels after hypoglycemia induction as a common metabolic signature providing new avenues for diagnostic and therapeutic approaches in HAAF."). Taurine is involved in so many physiological functions.

We thank the reviewer for their insights into the manuscript. Our results that show taurine levels are down, both in *Trpc5*-deficient mice and HAAF patients during hypoglycemia, are observational. We do not, and cannot, claim a causal link between the hypoglycemia and the taurine levels based on our observations alone. Nonetheless, it is an interesting starting point for follow-up studies. We are aware of the multiple physiological functions of taurine, and the various causes that can change its plasma levels. We have the following fragment in the discussion (page 22, line 671) of the manuscript to indicate this:

*"Among the metabolites analyzed, we specifically observed the decrease in taurine levels as a coincidence between *Trpc5* KO mice and HAAF patients under hypoglycemia. Taurine is one of the most abundant free amino acids in mammalian tissues (Ripps & Shen, 2012). It can be synthesized endogenously from cysteine or methionine, for which we did not observe any differences in the HAAF patients, arguing against a defect in taurine synthesis. Several preclinical studies emphasize cytoprotective effects of taurine and therapeutic ameliorations in experimental and human diabetes including neurotransmission (Sirdah, 2015), e.g. by attenuating hyperalgesia and abnormal Ca^{2+} signaling in sensory neurons (Li et al, 2005). The mechanisms that lead to changes in taurine plasma levels are very complex and may differ between *Trpc5* KO mice and in HAAF patients as taurine metabolism is influenced by many metabolic pathways (Hayes, 1988; Ripps & Shen, 2012)."*

Reviewer #3 (Remarks to the Author):

Remarks to be sent to the author:

In response to the comments made by this reviewer, the authors performed additional experiments. This reviewer finds the experimental data to be at a scientifically sufficient level. However, as noted below, the reviewer feels that the interpretation and the contextual flow should be corrected to more faithfully reflect the experimental data.

1) Based on the additional results, TRPC-5 was clearly shown to mediate a variety of stress-induced forms of adrenalin secretion from chromaffin cells. Thus, hypoglycemia is only one of these triggers. However, this important finding was described only in the Discussion section and did not impact the context of the manuscript.

The central dogma of the manuscript is the effects of *Trpc5*-deficiency during hypoglycemia. We describe our findings that the autonomic counterregulatory response in *Trpc5*-deficient mice is impaired due to a reduced adrenaline secretion from the chromaffin cells. While there are other triggers for adrenaline secretion, they do not impact the context of insulin-induced hypoglycemia.

The LPS-induced stress and histamine-induced stress experiments are merely a proof-of-concept that adrenaline release is also impaired in *Trpc5*-deficient mice models during other stress responses. The experiments on the adrenaline secretion during histamine- and LPS- induced stress happened in the context of the revision, due to the available mouse colonies and ethical approvals that were still running at that time, we had to use the *Trpc5*^{-/-} mice and the specific catecholaminergic *Trpc5* knockout mouse (*Trpc5*^{fx/0}; DBH-Cre⁺) which, is only introduced in figure 2 and 4 of the manuscript, while the hypoglycemia-induced stress responses are discussed in the context of figure 1H. At that point of the manuscript, those genotypes (*Trpc5*^{-/-} and *Trpc5*^{fx/0}; DBH-Cre⁺) were not introduced yet. Furthermore, *Trpc5* was not yet identified as the sole responsible channel for the phenotype. To not confuse the reader, we addressed these important results in the discussion part.

2) The overall results showed TRPC-5 to constitute the fundamental mechanism of adrenalin secretion in chromaffin cells, but not to CONTROL or REGULATE the mechanism specifically underlying hypoglycemia-induced adrenalin secretion. Therefore, the title and conclusion should be corrected so as to actually reflect the experimental results.

We agree that TRPC5 constitutes a fundamental mechanism for the adrenaline secretion. Adrenaline secretion is an important step in the counterREGULATION of severe hypoglycemia. Therefore, to the transitive properties of controlling, our title is supported by the experimental results that show that “TRPC5 controls the adrenaline-mediated counter regulation of hypoglycemia”.

We have modified the conclusion and summary in the revised manuscript to address this remark.

“Conclusion

Our study identifies for the first time TRPC5 channels in adrenal chromaffin cells as critical regulators of calcium-dependent adrenaline secretion. Reduced hypoglycemia-evoked adrenalin secretion, as observed in *Trpc5* KO deficient mice, contributes significantly to hypoglycemia-associated autonomic counter regulation. The similarities in sympatho-adrenal insufficiency between *Trpc5* KO mice and HAAF patients

identified for the first time represent a new avenue for the development of new TRPC5-associated diagnostic and treatment options for the long-term diabetic complication HAAF.”

3) The authors noted a striking parallel in sympatho-adrenal failure after the occurrence of hypoglycemia between *Trpc5*-KO mice and HAAF patients. However, since sympathetic activity during hypoglycemia is impaired in HAAF patients, it is unlikely that its pathological region resides in chromaffin cells. These findings suggest that the impaired process in HAAF patients is different from that mediated by TRPC 5. TRPC5 channel agonists thus do not appear to hold promise as HAAF therapies.

We have included several new passages in the discussion (lines 659 and 665) that address the functional difference between “TRPC5-dependent impaired adrenaline secretion from the chromaffin cells” and “HAAF due to impaired sympathetic activity”. In this context, TRPC5 agonist are potential symptomatic therapy for the reduced sympathetic activity.

*“Our results show a striking parallel in sympatho-adrenal failure after the occurrence of hypoglycemia between *Trpc5* KO mice and in HAAF patients. One might speculate that perturbations in the TRPC5 activity are involved in the pathophysiology of HAAF. However, the defect in autonomic counter regulation in *Trpc5* KO mice is not specifically triggered by induction of hypoglycemia but is also observed in other forms of stress and in HAAF patients, a reduced sympathetic activity is observed (Cryer, 2013), which is upstream of the chromaffin cells. Nevertheless, follow-up studies are needed to show more directly a contribution of the TRPC5-dependent mechanism to sympatho-adrenal failure. In this case, TRPC5 channel agonists such as englerin A (EA) or other more tolerable agonists, such as tonantzitolone, riluzole, or BTB (Minard et al, 2019) bear potential as symptomatic therapy, acting downstream of the impaired sympathetic activity, for the still-incurable long-term diabetic complication. “*

Dear Marc,

Congratulations on an excellent manuscript, I am pleased to inform you that your manuscript has been accepted for publication in the EMBO Journal. Thank you for your comprehensive response to the referee concerns and for providing detailed source data. It has been a pleasure to work with you to get this to the acceptance stage.

I will begin the final checks on your manuscript before submitting to the publisher next week. Once at the publisher, it will take about 3 weeks for your manuscript to be published online. As a reminder, the entire review process will be available to readers.

I will be in touch throughout the final editorial process until publication. In the meantime, I hope you find time to celebrate!

Warm wishes,
Kelly

Kelly M Anderson, PhD
Editor, The EMBO Journal
k.anderson@embojournal.org
